# Deletion of the *Candida albicans TLO* gene family using CRISPR-Cas9 mutagenesis allows characterisation of functional differences in α-, β- and γ- *TLO* gene function

Jessica Fletcher[1¤a], James O'Connor-Moneley[1], Dean Frawley[1¤b], Peter R. Flanagan[1¤c], Leenah Alaalm[1], Pilar Menendez-Manjon[2], Samuel Vega Estevez[3], Shane Hendricks[4], Andrew L. Woodruff[4], Alessia Buscaino[3], Matthew Z. Anderson[4], Derek J. Sullivan[1☯*], Gary P. Moran[1☯*]

1 Division of Oral Biosciences, Dublin Dental University Hospital, & University of Dublin, Trinity College Dublin, Dublin, Ireland, 2 Departamento de Enfermería I, Universidad del País Vasco, Bilbao, Spain, 3 School of Biosciences, University of Kent, Canterbury, United Kingdom, 4 Department of Microbiology, The Ohio State University, Columbus, Ohio, United States of America

☯ These authors contributed equally to this work.
¤a Current address: Dept. of Integrated Biology, University of Colorado, Denver, Colorado, United States of America
¤b Current address: Biology Department, Maynooth University, Maynooth, Co. Kildare, Ireland
¤c Current address: Dept. of Medical Microbiology, LabMed Directorate, St. James's Hospital, Dublin 8, Ireland
* derek.sullivan@dental.tcd.ie (DJS); gpmoran@dental.tcd.ie (GPM)

**Data Availability Statement:** Sequence data is available for download from the NCBI sequence

## Abstract

The *Candida albicans* genome contains between ten and fifteen distinct *TLO* genes that all encode a Med2 subunit of Mediator. In order to investigate the biological role of Med2/Tlo in *C. albicans* we deleted all fourteen *TLO* genes using CRISPR-Cas9 mutagenesis. ChIP-seq analysis showed that RNAP II localized to 55% fewer genes in the *tlo*Δ mutant strain compared to the parent, while RNA-seq analysis showed that the *tlo*Δ mutant exhibited differential expression of genes required for carbohydrate metabolism, stress responses, white-opaque switching and filamentous growth. Consequently, the *tlo*Δ mutant grows poorly in glucose- and galactose-containing media, is unable to grow as true hyphae, is more sensitive to oxidative stress and is less virulent in the wax worm infection model. Reintegration of genes representative of the α-, β- and γ-*TLO* clades resulted in the complementation of the mutant phenotypes, but to different degrees. *TLOα1* could restore phenotypes and gene expression patterns similar to wild-type and was the strongest activator of glycolytic and Tye7-regulated gene expression. In contrast, the two γ-*TLO* genes examined (*i.e.*, *TLOγ5 and TLOγ11*) had a far lower impact on complementing phenotypic and transcriptomic changes. Uniquely, expression of *TLOβ2* in the *tlo*Δ mutant stimulated filamentous growth in YEPD medium and this phenotype was enhanced when Tloβ2 expression was increased to levels far in excess of Med3. In contrast, expression of reintegrated *TLO* genes in a *tlo*Δ/ *med3*Δ double mutant background failed to restore any of the phenotypes tested, suggesting that complementation of these Tlo-regulated processes requires a functional Mediator tail module. Together, these data confirm the importance of Med2/Tlo in a wide range of *C.*

read archive. Genome sequences: BioProject no. PRJNA962819; RNA seq data: BioProject no. PRJNA962476; ChIP seq data: BioProject no. PRJNA962549.

**Funding:** GPM and DJS are funded by Science Foundation Ireland (SFI, No. 19/FFP/6422) to study the function of the *Candida albicans TLO* gene family. MZA is funded by an NSF Career Award (No. 2046863) and ALW is funded by an NIH F31 (No. 1AI167576). The funders did not play any role in study design, data collection, analysis or decision to publish.

**Competing interests:** The authors have declared that no competing interests exist.

*albicans* cellular activities and demonstrate functional diversity within the gene family which may contribute to the success of this yeast as a coloniser and pathogen of humans.

## Author summary

Fungi are an important cause of infectious disease in humans. One of the most potent causes of fungal infections is the single-celled yeast species, *Candida albicans*, which usually lives in balance as part of the microbial communities that inhabit the gastrointestinal and vaginal tract. However, under certain conditions, for example in immunocompromised hosts, the yeast can overgrow and cause a range of infections ranging from superficial to life-threatening diseases. Our understanding of this organism and how it causes disease is still rudimentary. One of the most unique aspects of the genome of this species is the presence of an expanded family of fourteen *TLO* genes that encode the Med2 component of Mediator, a multi-peptide complex involved in the transcription of genes. Using CRISPR-Cas9-mediated mutagenesis we have successfully deleted all fourteen *C. albicans TLO* genes. Molecular and phenotypic analysis of the mutant and strains expressing representative *TLO* genes, directly confirms that the *TLO/MED2* gene family is required for a wide range of important cell functions, including carbohydrate metabolism, cell morphology, stress tolerance and virulence, and that there is functional diversity within the proteins encoded by individual *TLO* genes.

## Introduction

Fungi are an important cause of morbidity and mortality in humans, and there is an urgent need to develop novel therapies to treat fungal infections [1–3]. The yeast species *Candida albicans* is a constituent of the microbiome in the human gastrointestinal tract, the oral cavity and the vagina [4,5]. Although its growth is kept under control by the host immune response, if there is a host immunodeficiency (*e.g.*, caused by HIV infection or immunosuppression therapy) or if there is an imbalance in the mucosal microbiota (*e.g.*, due to antibacterial treatment), *C. albicans* can overgrow its natural niche and lead to the development of superficial and/or systemic infections [4,6].

The *Candida* genus is comprised of dozens of phylogenetically diverse species. However, *C. albicans* remains the most clinically relevant yeast species [4,6–8], which was highlighted recently by its inclusion in the "Critical Group" of the World Health Organization's fungal priority pathogens list [7,8]. Comparative genomic analysis has been used to investigate the underlying basis for the differential virulence evident among *Candida* species [9,10]. These analyses suggest that gene family expansions have played an important role in the evolution of pathogenicity in *C. albicans*. Gene duplication events may have facilitated phenotypic plasticity through neo-specialization, allowing some paralogs to be differentially expressed during infection (*e.g.*, the Sap family of proteinases) or to acquire specific functions (*e.g.*, the invasin Als3) [11,12]. One of the most prominent gene family differences between *C. albicans* and other *Candida* species is a paralogous expansion of the *TLO* (te̲lo̲mere-associated) gene family which encodes the Med2 component of the master transcriptional regulator Mediator [10,13]. Mediator is a multiprotein complex that bridges interaction between RNA polymerase II (RNAP II) and a wide range of transcription factors, as well as playing a role in preinitiation complex formation and chromatin remodelling [14,15]. Typically, yeast Mediator complexes are

composed of 25 polypeptides, grouped into four modules (i.e., head, middle, tail and kinase modules), The Med2 subunit is found in the tail module together with the Med15 and Med3 subunits [14]. The role of Mediator in fungal biology is relatively poorly understood, particularly in *Candida* species. Half of all fungal Mediator subunits are encoded by essential genes, and these have about 20–30% amino acid identity with their homologs in mammals [13,16–18]. Deletion of genes encoding specific Mediator subunits (e.g. *MED7* and *MED31*) demonstrates that Mediator is involved in a wide range of *C. albicans* biological processes, including expression of virulence genes, hypha formation, white-opaque switching, mating and glycolysis [19–21].

The genomes of most fungal species encode a single *MED2* gene, however *C. albicans* clinical isolates contain 10 to 15 *TLO/MED2* genes, mostly located at subtelomeric regions. The closely related species *C. dubliniensis* is also unusual in that it encodes two *TLO* genes [10]. It has been suggested that the *TLO* gene family is rapidly evolving, in terms of the copy number and location of the *TLO* genes in the *C. albicans* genome [22] and that the family may have expanded from a single ancestral *TLO* gene (possibly *TLOβ2*) through subtelomeric recombination events [10,23,24]. While all Tlo proteins contain a conserved N-terminal Med2 domain that allows integration into the Mediator complex, sequence divergence in the C-terminal half of the protein separates them into three distinct groups (referred to as the α-, β- and γ- clades) [25,26]. In the reference *C. albicans* strain SC5314, there are seven members of the γ-clade, which are the shortest *TLO* genes at approximately 525 bp [25,26] and the only clade that has been shown to undergo alternative splicing. The α-clade of *TLO*s (comprised of five members) measure between 675 and 750 bp. The β-clade contains one single *TLO*, *TLOβ2*, which is 822 bp in length. Genes within the same *TLO* clade share up to 97% nucleotide sequence homology while genes from different clades are approximately 82% identical. The different clades of Tlo proteins display differing patterns of cellular localisation, with most Tlos containing a nuclear localisation sequence that has been proposed to direct the proteins to the nucleus, while proteins from unspliced *TLOγ* transcripts have been shown to localise to both the nucleus and the mitochondria [25,26]. Protein expression analysis has also revealed that, unlike in other species, in *C. albicans* there is a large population of "free" Tlo protein in excess to that bound by Mediator [13]. It has also been shown that artificially creating an excess of "free" Tlo proteins in *C. dubliniensis* (by overexpressing *CdTLO2*, but not *CdTLO1*) resulted in an increased ability of this species to filament, potentially demonstrating a role for free Tlo in the virulence of *C. dubliniensis* [16].

Given the uniqueness of the expansion of the *TLO* gene family in *C. albicans* it has previously been suggested that the potential diverse pool of free Tlos and Mediator complexes containing specific Tlo proteins may play a role in the relative success of this species as an opportunistic pathogen [25,27–29]. Deletion of the two *TLO* genes in the genome of the closely related species *C. dubliniensis* supports the hypothesis that *TLO* genes are required for pathogenicity [10]. At the same time, expression of clade-specific *C. albicans TLO* genes in the *C. dubliniensis TLO* null mutant and induced overexpression of individual *TLO* genes in *C. albicans* produced differential phenotypic outcomes, suggesting the potential interplay between expression and encoded genetic effects of *TLO* genes [27,30]

Definitive evidence confirming the importance and functional diversity of the *TLO* gene family in *C. albicans* has been thwarted by the inherent difficulty in manipulating such a large family of genes. Here we describe the application of CRISPR-Cas9 technology to delete the entire complement of fourteen *TLO* genes in a reference *C. albicans* strain. This allowed us to investigate the effects of reintroducing single *TLO* genes by comparing the phenotypes and transcriptional profiles of the *tloΔ* mutant and strains reconstituted with a representative *TLO* gene from each of the three *TLO* α-, β- and γ-gene groups.

## Results

### CRISPR-Cas9 deletion of *TLOs* in *Candida albicans*

To unequivocally identify the collective function of the *C. albicans TLO* gene family, we used a CRISPR-Cas9 approach to delete all fourteen *TLO* genes present in the reference strain AHY940 (S1A Fig). In parallel, we also used CRISPR-Cas9 gene editing to delete the *MED3* gene encoding the Mediator Med3 tail module component which is required for Tlo incorporation into Mediator complex. Successful deletion of each of the 28 *TLO* alleles was initially demonstrated using PCR primers targeting each *TLO* locus (S2 Table and S1B Fig) and subsequently confirmed by whole-genome sequencing of two independently derived mutants, CC10 and CC16. The *C. albicans* genome has been shown to be plastic and due to the extensive degree of double stranded break repair involved in the generation of the *tloΔ* null mutants, we expected that this could likely result in the generation of chromosomal rearrangements [31]. Therefore, to investigate this we used a combination of long and short read genome sequencing data and pulsed-field gel electrophoresis (PFGE) to characterise the chromosomal structure of the two independent *tloΔ* null mutants. Unexpectedly, sequence analysis revealed that the AHY940 parental strain used to generate the *tloΔ* null mutants was trisomic for chromosome 5 (ABB; S2 and S3 Figs). Importantly, both *tloΔ* mutants selected for subsequent analysis were completely diploid, having resolved the trisomic Chr5 karyotype to become disomic for Chr5, retaining one of each homolog A and B. Using Illumina short read sequencing, we examined heterozygosity in each *tloΔ* strain compared to the parental CRISPR competent strain. In CC10 a single loss of heterozygosity (LOH) event was detected on the extreme left end of chromosome 6 (Chr6) approximately 55 kb long. The CC16 mutant did not show any evidence of new areas with LOH compared to the parent.

Analysis of chromosome architecture using a combination of PFGE and long read sequencing revealed the presence of novel chromosomes in both *tloΔ* strains. *TLO* null strain CC16 had rearrangements in both homologs of Chr1. One Chr1 homolog exhibited a truncation of the left arm (Chr1L) located in the 3' coding sequence of *TLOα34*, producing a new telomere for the left arm within the *TLOα34* coding sequence (S3 Fig). The Chr1L fragment from this truncated Chr1 homolog had fused to one Chr7 homolog at a homologous Rho LTR sequence immediately downstream of *TLOα34* and centromeric to *TLOγ16* (S3 Fig). Although the second Chr1 homolog was entirely intact, it was found to contain an inversion beginning at the *TLOα34* locus through Chr1R, including the centromere (S3 Fig). This inversion resulted from recombination between the Rho long terminal repeat (LTR) telomeric of *TLOα34* on Chr1L fragment and the Rho LTR located centromeric of *TLOγ4* on Chr1R. The presence of these Chr 1 variants was inferred from CHEF gels of whole chromosomes (S4 Fig). The *tloΔ* strain CC10 contained more extensive rearrangements involving multiple chromosomes as described in S1 Text and S3 Fig.

Chromosomal rearrangements present in both the CC16 and CC10 *tloΔ* null mutants required careful consideration when assigning mutant phenotypes. As these mutants display distinct chromosome rearrangements, phenotypes observed in both *tloΔ* mutants will likely be the result of the *TLO* gene deletion rather than chromosomal rearrangement events. Thus, all mutant phenotypes described below were confirmed in both *tloΔ* null mutants CC10 and CC16. Detailed molecular analyses (e.g. RNA-seq) were conducted using the CC16 mutant as it has undergone fewer rearrangements than any of the mutants examined. We also generated a Chr5 disomic (AB) derivative of the wildtype trisomic parent AHY940, termed MAY1244 (described in S1 Text and S2 Fig) for comparisons to the *tloΔ* mutants that are also fully diploid. MAY1244 is phenotypically identical to AHY940 in the assays described here and was used in genotypic analyses.

## The *tloΔ* mutant is transcriptionally distinct and maintains less RNA polymerase II occupancy

RNA sequencing analysis was performed to compare the transcriptomes of the parental WT strains (AHY940 and MAY1244) and the *tloΔ* mutant CC16, grown to mid-exponential phase in YEPD at 37°C. The *tloΔ* mutant exhibited a transcriptome significantly different from WT (n = 999 genes 2-fold different from MAY1244, FDR q < 0.05, Fig 1A and S3 Table). Comparison of the mutant transcriptome with AHY940 and MAY1244 identified consistent differences including reduced expression of genes required for carbohydrate metabolism, amino acid metabolism, white phase genes, stress response genes and genes regulated during hyphal growth (Fig 1B). Genes shown to interact with the carbohydrate metabolism regulating transcription factors Gal4 and Tye7 showed significantly greater expression in the wild-type strains, in addition to genes repressed by the stress-response regulator Sko1. The *tloΔ* mutant also exhibited significantly higher expression of genes involved in ribosome biogenesis (Fig 1B). Many of these changes (e.g. reduced carbohydrate metabolism, increased ribosome biogenesis, increased

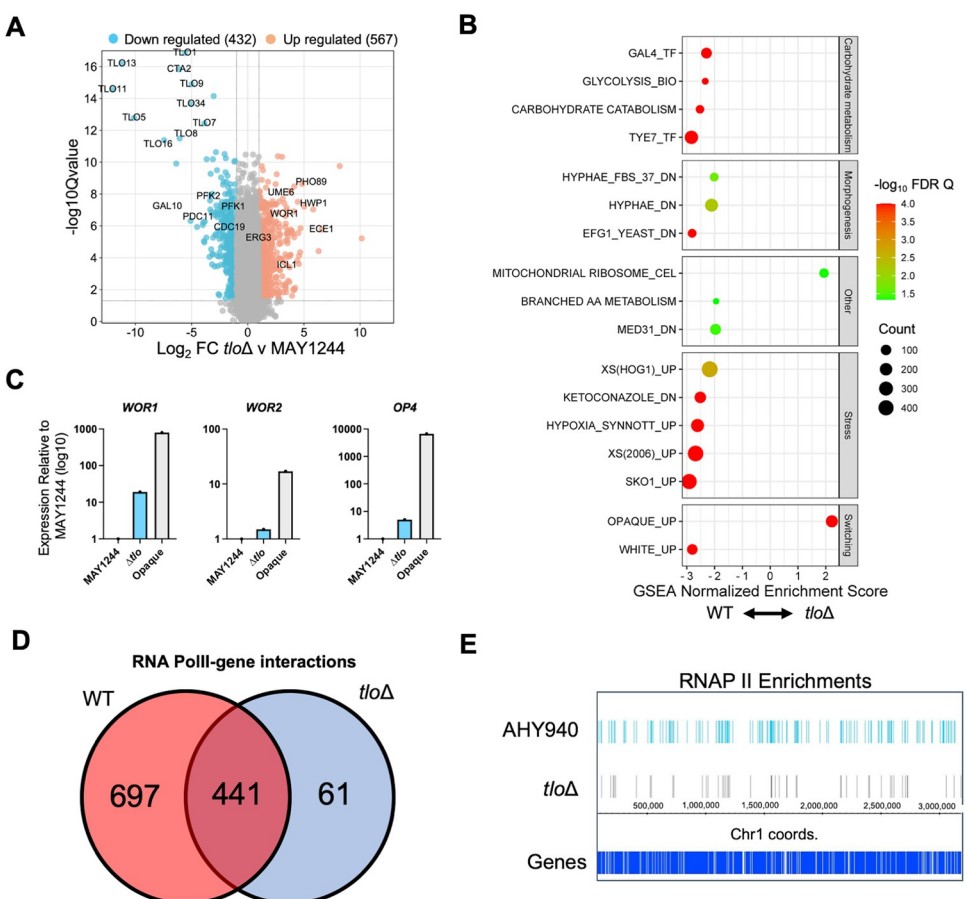

**Fig 1. Analysis of gene expression in the *tloΔ* mutant CC16.** (A) Differential gene expression in *tloΔ* versus MAY1244 following RNA-seq in YEPD medium at 37°C. Highlighted genes exhibit 2-fold difference in gene expression (FDR q value <0.05). (B) GSEA of RNA-seq data in (A) showing categories of genes enriched in the transcriptomes of MAY1244 or the *tloΔ* mutant. Categories with FDR q value <0.05 are indicated with GSEA enrichment score (positive = *tloΔ* enriched, negative = WT enriched) and symbol diameter related to the number of genes (count). (C) Expression of *WOR1*, *WOR2* and *OP4* in *tloΔ* and opaque cells of RBY177 relative to white cells of MAY1244. (D) Comparison of the number of anti-RNAP II ChIP enriched regions in AHY940 and the *tloΔ* mutant. (E) Distribution of anti-RNAP II ChIP enriched regions across chromosome 1 in AHY940 and the *tloΔ* mutant.

hypha-specific gene expression) have been described in other Mediator mutants, including the *med7Δ* mutant [20] and the *C. dubliniensis med3Δ* and *tloΔ* mutants [28].

In comparison to these previous Mediator studies, a unique feature of the *C. albicans tloΔ* mutant transcriptome was increased expression of genes normally expressed in opaque cells of *C. albicans*. Despite this pattern of gene expression, the *tloΔ* mutant did not express an obvious opaque cell morphology, even when grown under conditions that favoured the switch from the white to opaque cell morphology, likely due to a non-functional Mediator tail [21]. We hypothesized that expression of factors that promote opaque cell growth may not be expressed at high enough levels to trigger the opaque cell transition. qRT-PCR analysis of *WOR1*, *WOR2* and *OP4* expression levels in the *tloΔ* mutant supported this hypothesis and showed that *ACT1*-normalized opaque phase gene expression in the *tloΔ* mutant was at least an order of magnitude lower than that observed in opaque phase cells of strain RBY1177 (*MTL*a) (Fig 1C).

As the Mediator complex is required for recruitment and regulation of RNAP II, we next used an anti-RNAP II antibody in ChIP-seq experiments to investigate if genome-wide localization of RNAP II differed in the parent strain (AHY940) and the *tloΔ* mutant. RNAP II was found to be much less widely distributed in the *tloΔ* mutant strain compared to the parent, indicated by the reduced number of significant (FDR q <0.05) ChIP-enriched regions identified by MACS2 in the *tloΔ* mutant (n = 502) compared to the parent strain (n = 1,138) (Fig 1D and 1E). The expression of genes that only interacted with RNAP II in the WT strain (n = 697, Fig 1D) was found to be significantly higher in WT compared to the *tloΔ* mutant strain (matched Wilcoxon p < 0.001, S5 Fig).

## *tloΔ* mutants display altered cellular morphologies, reduced growth rates and increased stress sensitivity

A range of phenotypic tests was used to investigate the effect of deleting the entire repertoire of *TLO* genes in *C. albicans*. The colony morphology of the two *tloΔ* mutant strains tested was similar to that of the WT strain when grown on solid YEPD agar at 37˚C (Fig 2A). However, at the cellular level, while parental strains displayed a typical yeast morphology, the *tloΔ* mutants grew predominantly as pseudohypha-like chains of cells (Fig 2A), possibly suggestive of a cell separation defect. We observed similar cell morphologies in a *med3Δ* mutant background, although *med3Δ* colonies were rougher in texture (Fig 2A). The *tloΔ* and *med3Δ* mutants also showed altered morphology on solid Spider medium characterised by the absence of a hyphal fringe typical of WT colonies (S6A Fig). In liquid hypha-inducing medium (*i.e.*, 10% serum at 37˚C), the *tloΔ* and *med3Δ* mutants exhibited an inability to form true hyphae following incubation for up to 4 h (Fig 2B). Growth rates in YEP-glucose and YEP-galactose were significantly reduced in the *tloΔ* mutants, exhibiting growth rates not significantly different to the *med3Δ* mutant (Fig 2C). Deletion of the *TLO* gene family also increased sensitivity to oxidative stress (both $H_2O_2$ and tert-Butyl hydroperoxide [tBOOH]) to a level similar to that observed in the *med3Δ* mutant (Fig 2D). The *tloΔ* mutants and the *med3Δ* mutant also exhibited increased sensitivity to cell wall stress induced by Congo Red (Fig 2E). The ability of the *tloΔ* mutants and the *med3Δ* mutant to form biofilm on plastic surfaces was also greatly reduced relative to AHY940 (S6B Fig). We also assessed virulence in the *G. mellonella* infection model and found that the *tloΔ* mutants exhibited significantly reduced larval death compared to AHY940, similar to the *med3Δ* mutant (Fig 2F).

## Complementation of the *tloΔ* mutant with *TLOα1*, *TLOβ2* and *TLOγ11*

In order to assess the contributions of individual *TLO* genes to phenotypes displayed by the *tloΔ* mutant, we complemented the CC16 *tloΔ* mutant with clade-representative alpha

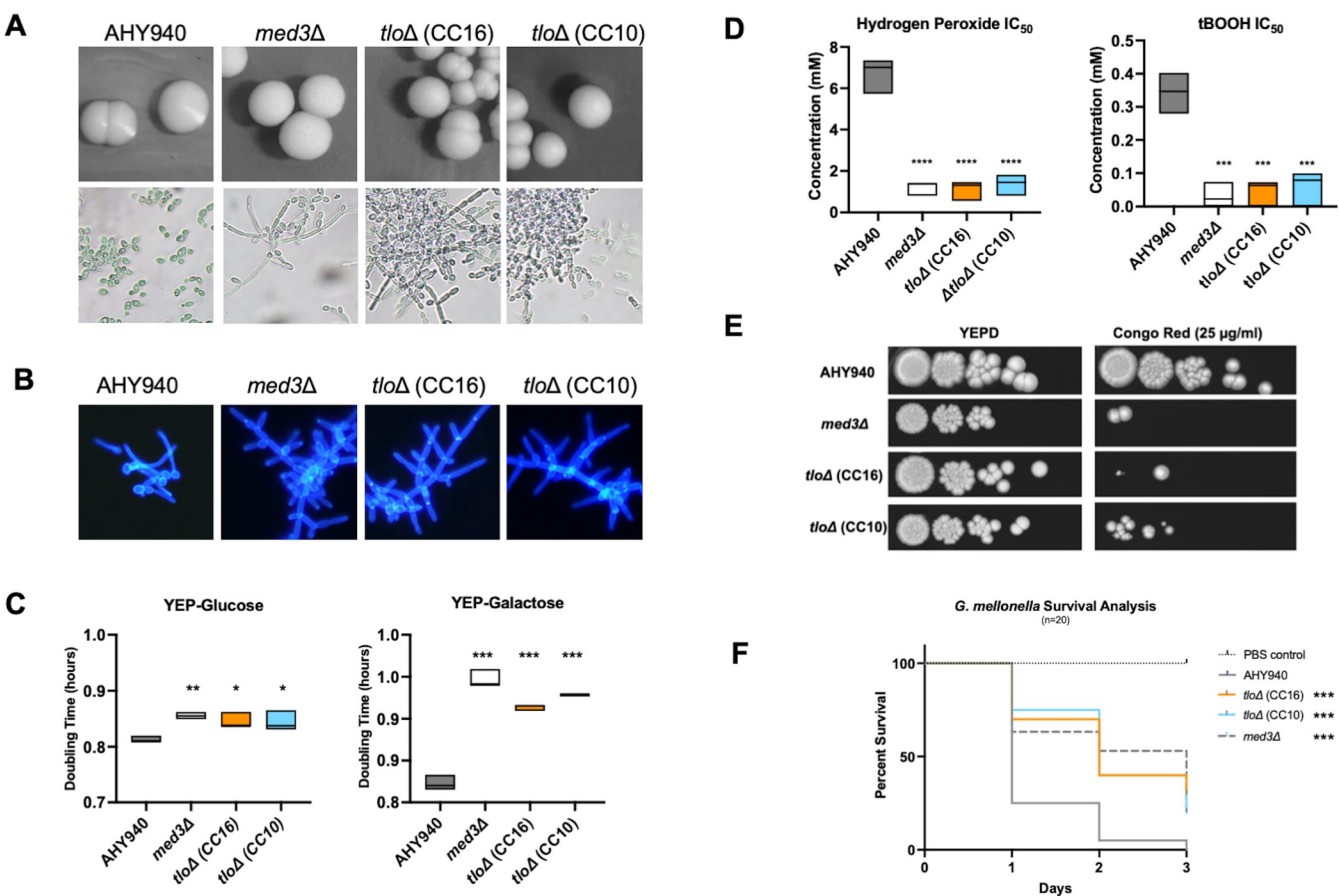

**Fig 2. Phenotypic analysis of *tloΔ* (CC10 and CC16) and *med3Δ* mutants relative to WT AHY940.** (A) Colony and cellular morphology on YEPD medium at 37˚C. (B) Cellular morphology following 2h incubation in YEPD + 10% serum at 37˚C. (C) Growth rates (doubling time, h) in YEP-2% glucose and YEP-2% galactose. Significant differences from WT (one-way ANOVA) are indicated as follows: *** = p < 0.001; ** = p <0.01, * = p < 0.05). (D) IC$_{50}$ values of hydrogen peroxide and tBOOH respectively. (E) Susceptibility to Congo Red in spot plate assay. (F) Survival of *G. mellonella* larvae following infection with the indicated strains, *** = p <0.01 in log-rank (Mantel-Cox) test.

(*TLOα1*), beta (*TLOβ2*) or gamma (*TLOγ11*) genes. Each reintroduced *TLO* gene was driven by a relatively weak and a strong constitutive promoter (i.e., $P_{TET}$ and enolase $P_{ENO1}$ promoters, respectively) to facilitate the investigation of gene/clade- and expression-associated phenotypes (S7 Fig). These promoters produced mRNA expression levels either equivalent to ($P_{ENO1}$) or below ($P_{TET}$) those from the native genes (Fig 3A). Expression of Tloα1 and Tloβ2 HA-tagged proteins could be detected by Western blotting from both promoters using anti-HA antibodies and higher levels were detected from the enolase promoter-expressed gene (Fig 3B). Analysis of Tlo band intensity relative to the RNAP II loading control showed that the levels of Tloα1 and Tloβ2 HA-tagged proteins were equivalent when expressed by each of the promoters (i.e., the levels of $P_{TET}$-Tloα1 were similar to $P_{TET}$-Tloβ2). While *TLOγ11* mRNA was detectable by qRT-PCR, HA-tagged Tloγ11 (both C- and N-terminally tagged variants) could not be detected by Western blot. Expression of a second HA-tagged gamma Tlo (Tloγ5) was similarly undetectable. Increased loading or treatment of protein extracts with a proteosome inhibitor prior to immunoblotting also did not lead to Tloγ protein detection. However, following immunoprecipitation (IP) of $P_{ENO}$-*TLOγ11* strain protein extracts with anti-HA antibody and tandem mass-spectrometry (LC-MS/MS), we could detect Tloγ11 peptides (see below), indicating that the Tloγ11-HA protein was translated.

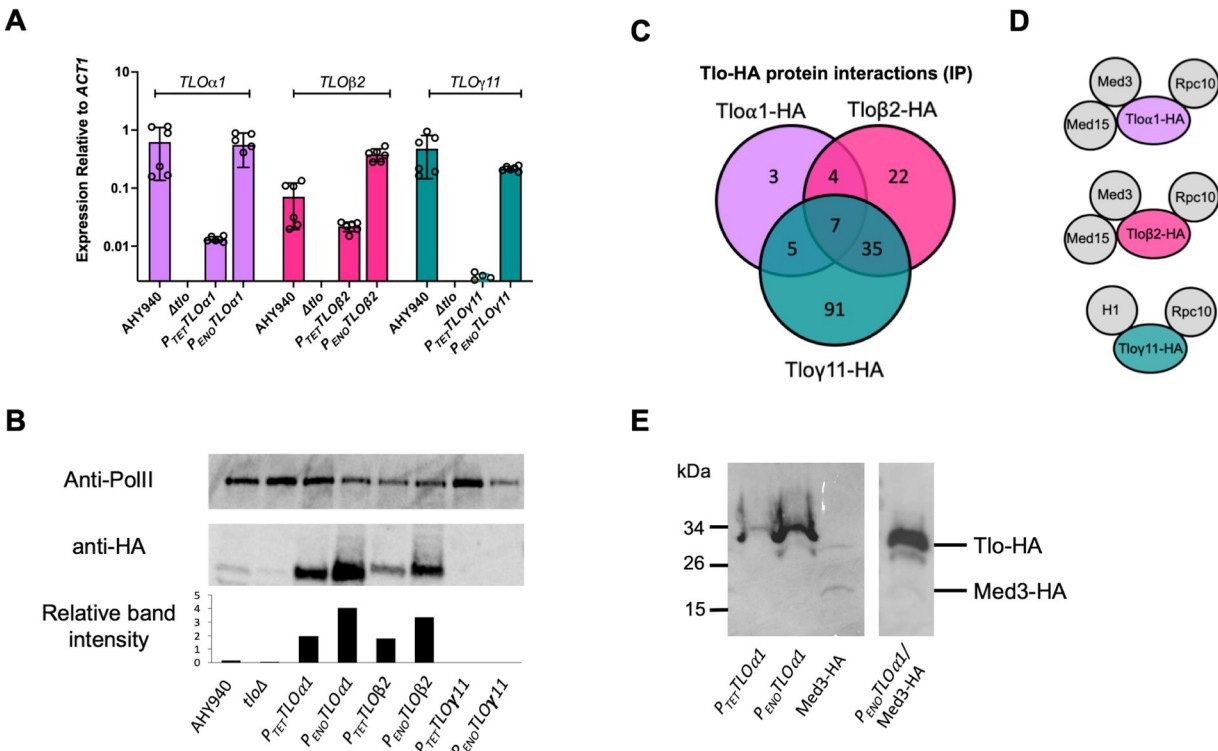

**Fig 3. Complementation of the *tlo*Δ mutant with *TLOα1*, *TLOβ2* and *TLOγ11*.** (A) qRT-PCR analysis of expression of *TLOα1*, *TLOβ2* and *TLOγ11* in WT AHY940, *tlo*Δ mutant and *tlo*Δ mutant strain complemented with $P_{TET}$ or $P_{ENO1}$ expressed genes. (B) Western blot showing expression of HA-tagged $P_{TET}$ or $P_{ENO1}$ expressed genes in the complemented *Δtlo* mutant. Relative band intensity graph indicates Tlo band intensity relative to RNAP II band intensity following quantification with GelAnalyzer 19.1 (C) Venn diagram showing overlapping proteins identified in triplicate anti-HA IP experiments with protein extracts from Tloα1-HA, Tloβ2-HA or Tloγ11-HA expressing strains. (D) Schematic showing Mediator and RNAP II interactions identified in Tloα1-HA, Tloβ2-HA or Tloγ11-HA IP experiments. (E) Western blot analysis showing expression of HA-tagged Tloα1 relative to Med3-HA in a singly tagged strain (left panel) and in a dual Tloα1/Med3 HA-tagged strain (right panel).

In order to determine whether the HA-tagged Tloα1, Tloβ2 and Tloγ11 proteins were incorporated into the Mediator complex, we next conducted IP experiments followed by LC-MS/MS. IP with anti-HA antibody was shown to efficiently isolate the majority of cellular Tlo protein (S8 Fig). Three independent replicates were performed for each Tlo and in the case of Tloα1, 19 potential interacting proteins were identified across all three IP experiments (Fig 3C and S4 Table). IP of Tloβ2 identified 68 proteins, and, in the case of Tloγ11, 138 proteins were identified (Fig 3C and S5 and S6 Tables). Each Tlo protein was identified in the corresponding IP experiment, including Tloγ11. Seven common interacting partners were identified for all three Tlo proteins including the core subunit of RNA polymerase Rpc10, translation elongation factor Tef1 and translation initiation factor Tif3. Tloα1 and Tloβ2 were found to interact with the other subunits of Mediator tail, Med3 and Med15, while these were not detected in any of the Tloγ11 replicates (Fig 3D). Tloγ11 was found to uniquely interact with histone H1 among 74 additional unique proteins from diverse cellular locations (S6 Table and Fig 3D). Transcriptional activators were absent among Tlo-associated proteins, suggesting either indirect or weak interactions with Tlo or that they interact more closely with the other Mediator tail subunits (e.g., Med3 or Med15).

Wild-type *C. albicans* has been shown to express a population of excess "Mediator-free" Tlo which has been hypothesized to function independently of the Mediator complex [13]. Therefore we wished to determine whether the $P_{TET}$ and $P_{ENO1}$ promoter-expressed Tloα1 and

Tloβ2 proteins in the reintegrant strains were present at levels equivalent to, or in excess of, the Mediator complex. We compared expression of HA-tagged Med3 to Tloα1-HA (both $P_{TET}$ and $P_{ENO1}$) which showed that Tloα1 expression was greater than Med3-HA expression (at least 5-fold greater; Fig 3E). We also generated a strain that expressed both $P_{ENO1}$ expressed Tloα1-HA and Med3-HA. Comparison of Med3-HA and Tloα1-HA expression in this dual tagged strain showed at least a 30-fold excess of Tloα1 relative to Med3 (Fig 3E), suggesting that there may be a population of Mediator-free Tloα1-HA.

## Complementation of the *tloΔ* mutant with *TLOα1*, *TLOβ2* or *TLOγ11* restores different patterns of gene expression

Global gene expression analysis showed that complementation of the *tloΔ* mutant with $P_{ENO1}$-*TLOα1* or -*TLOβ2* resulted in a similar magnitude of differentially expressed genes with 367 and 311 genes significantly upregulated (2-fold; FDR q <0.05) and 491 and 463 genes down-regulated (2-fold; FDR q <0.05), respectively (Fig 4A and S7 and S8 Tables). Restoration of *TLOγ11* had a far lower impact on the transcriptome, with only 88 up-regulated and 133 down-regulated genes (Fig 4A and S9 Table).

Next, GSEA was used to analyse these differentially regulated gene lists (S10–S13 Table). Complementation of the mutant with $P_{ENO1}$-*TLOα1* or -*TLOβ2* reversed the misexpression of opaque phase genes in the *tloΔ* mutant (e.g. *WOR1*, *OP4*) and enhanced the expression of white phase genes (Fig 4B). *TLOα1* or *TLOβ2* also restored the reduced expression of genes involved in glycolysis (e.g., *TYE7*) while repressing the up-regulation of amino acid biosyn-thetic genes and ketoconazole induced genes upregulated in the mutant background (Fig 4B). *TLOγ11* only weakly affected these processes and did not significantly alter the expression of any gene sets in GSEA experiments (Fig 4B).

The enhanced expression of the hypha-specific genes *ECE1*, *HYR1*, *ALS3* and *HWP1* observed in the *tloΔ* mutant was strongly suppressed by reintroduction of a *TLOα* member. Complementation with $P_{TET}$-*TLOα1* or $P_{ENO}$-*TLOα1* elicited a ~10-fold reduction in hypha-specific gene expression (Fig 4C). Expression of *TLOβ2* from the $P_{TET}$ promoter also repressed the expression of several hypha-specific genes relative to the *tloΔ* mutant. However increasing *TLOβ2* expression from the stronger $P_{ENO1}$-*TLOβ2* promoter resulted in increased expression of most hypha-specific genes when compared to the $P_{TET}$-*TLOβ2* strain or any of the *TLOα1* expressing strains (Fig 4C).

We next compared the $P_{ENO1}$-*TLOα1*, -*TLOβ2* and -*TLOγ11* reintegrant transcript profiles to a WT *C. albicans* (MAY1244) which showed that $P_{ENO1}$-*TLOα1* and -*TLOβ2* more closely resembled WT compared to $P_{ENO1}$- *TLOγ11* (S9 Fig). Although a wild-type transcriptome was largely restored by $P_{ENO1}$-*TLOα1* and *TLOβ2*, we could still observe enhanced expression (> 2-fold) of a number of genes which included *HWP1*, *ECE1* and in the case of *TLOβ2*, *WOR1*, relative to WT *C. albicans* (S9 Fig).

## Tlo proteins have common and distinct ChIP enrichment patterns

Chromatin immunoprecipitation of HA-tagged Tloα1, Tloβ2 and Tloγ11 proteins in the com-plemented CC16 *tloΔ* mutant strain was used to investigate the binding patterns of each *TLO* clade representative. This analysis identified regions of significant enrichment (FDR q <0.05) for each tagged Tlo protein in the absence of any other competing Tlo proteins (Fig 5A). Simi-lar to our *C. dubliniensis* Tlo ChIP analysis [28], we found that the Tloα1, Tloβ2 and Tloγ11 proteins preferentially localised to ORF bodies more commonly than to upstream regions (S10A Fig). More than 60% of Tlo-gene interactions were solely localised to the ORF body with the remaining peaks overlapping with the promoter region (-1 to -500) (S10B Fig). Some

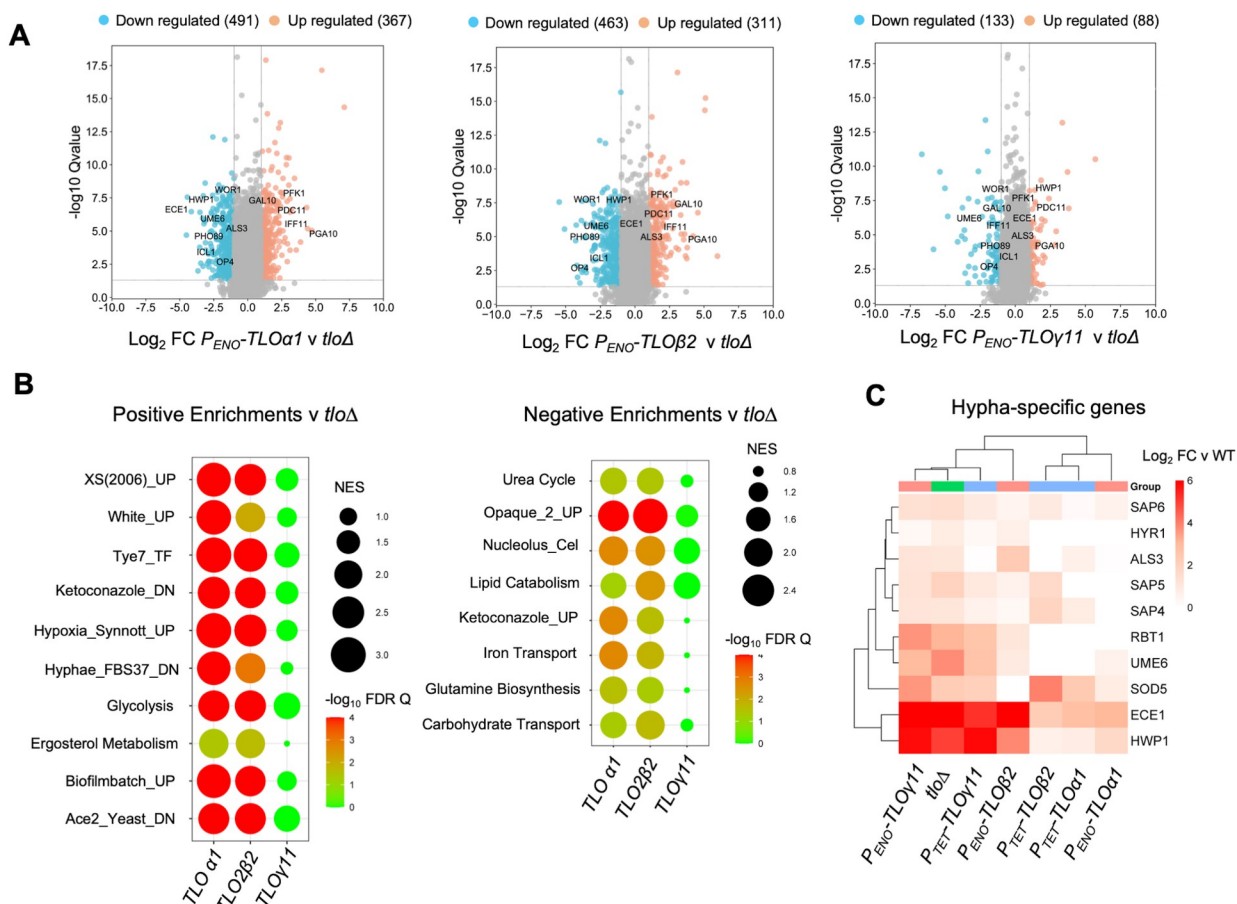

**Fig 4. Gene expression in $P_{ENO}$-*TLO* complemented strains.** (A) Volcano plots with spots representing genes with 2-fold or more change in gene expression (blue decreased; orange increased) relative to the *tloΔ* mutant (FDR q <0.05). (B) GSEA of gene expression in $P_{ENO}$-*TLO* complemented strains showing categories with higher expression in the $P_{ENO}$-*TLO* complemented strains (Positive enrichments) or reduced expression in $P_{ENO}$-*TLO* complemented strains (Negative enrichments). Circle size reflects the GSEA normalised enrichment score (NES) and color indicates significance (all FDR q < 0.05). (C) Heatmap showing expression of hypha-specific genes in the indicated strains relative to WT MAY1244.

Tlo enrichment was also found at repetitive regions of the genome, such as the MRS and centromeres (S10A Fig). Tloα1 had the lowest number of significantly bound genes (552; S14 Table) compared to the Tloβ2 (2,329; S15 Table) and Tloγ11 (2,521; S16 Table) data sets. All three Tlo proteins bound a core group of 275 genes, of which 42 corresponded to tRNAs. Gene ontology (GO) analysis of the remaining core protein encoding genes identified several GO categories including metabolic processes, filamentous growth and response to stress (S10C Fig). Genes of interest that exhibited Tlo enrichment included the transcriptional regulators *TYE7* and *GAL4*, previously highlighted in RNA-seq experiments (Fig 5B).

In order to determine if the presence of Tlo at an ORF correlated with expression, we examined the expression of the Tloα1, Tloβ2 and Tloγ11 interacting genes in the RNA seq data sets generated for each *TLO* reintegrant. In each data set, Tlo-interacting genes were significantly more highly expressed than non-Tlo interacting genes. We also observed that the normalised raw expression of Tloα1 and Tloβ2 interacting genes was significantly higher than for Tloγ11 interacting genes (Fig 5C).

We next carried out GSEA analysis of the Tlo interacting gene sets using the differential gene expression data for each Tlo reintegrant versus *tloΔ* (Fig 4) to infer the direct targets of

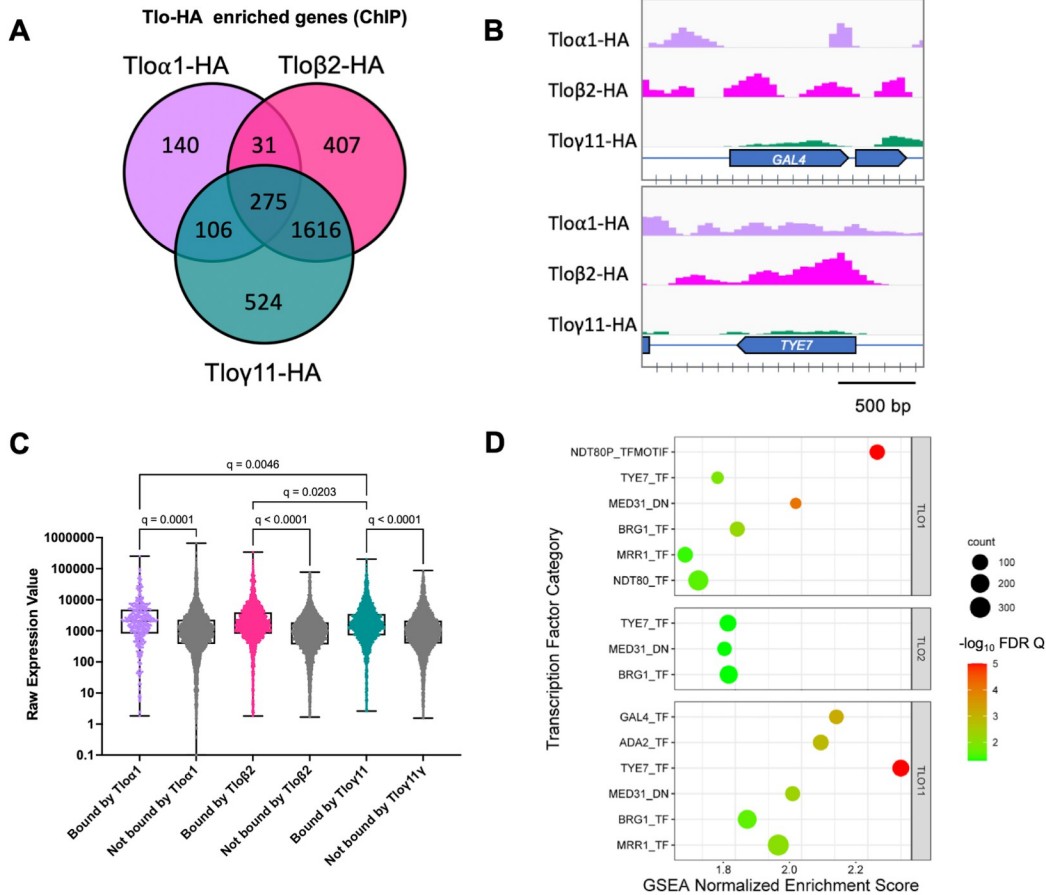

**Fig 5. Results of ChIP-seq experiments with HA-tagged Tlo proteins.** (A) Total numbers of genes exhibiting significant enrichment (FDR q < 0.05) following ChIP with HA-tagged Tloα1, Tloβ2 and Tloγ11 proteins. (B) ChIP enrichment patterns for Tloα1, Tloβ2 and Tloγ11 at the *TYE7* and *GAL4* ORFs. (C) Expression of genes bound by Tloα1, Tloβ2 and Tloγ11 compared to genes not bound by Tlo in RNA seq experiments of each *TLO* reintegrant expressed in *tlo*Δ. Expression refers to raw normalised expression values generated using Stand NGS. Statistical analysis (q values) indicate FDR corrected p values generated by ANOVA. (D) Results of GSEA analysis of the Tlo-bound gene sets of Tloα1, Tloβ2 and Tloγ11 (FDR q ≤ 0.05).

Tlo regulation. Using GSEA, we found the set of Tlo target genes was significantly associated with specific transcription factor regulons (FDR q ≤ 0.05; Fig 5D). Each of the Tlo-enriched gene sets included genes that interact with Tye7 and Brg1 as well as containing genes previously shown to be regulated by Mediator (Med31_DN). Tloα1 also showed significant association with Ndt80 while Tloγ11 was also significantly associated with Gal4 and Ada2.

## Complementation of the *tlo*Δ mutant with *TLOα1*, *TLOβ2* or *TLOγ11* has specific effects on morphology and biofilm formation

In order to investigate if the proteins expressed by *TLOα1*, *TLOβ2* or *TLOγ11* regulate different processes, we compared the phenotypes of each complemented mutant. Expression of *TLOα1* in the *tlo*Δ mutant background (under control of either the $P_{TET}$ or $P_{ENO1}$) reversed the pseudohypha-like phenotype of the CC16 *tlo*Δ mutant background and restored normal yeast cell morphology (Fig 6A). Restoration of non-filamentous morphology by complementation with *TLOα1* was also confirmed in the second *tlo*Δ mutant background, CC10 (S11 Fig). Expression of *TLOβ2* in the *tlo*Δ mutant background resulted in a mixed population of yeast

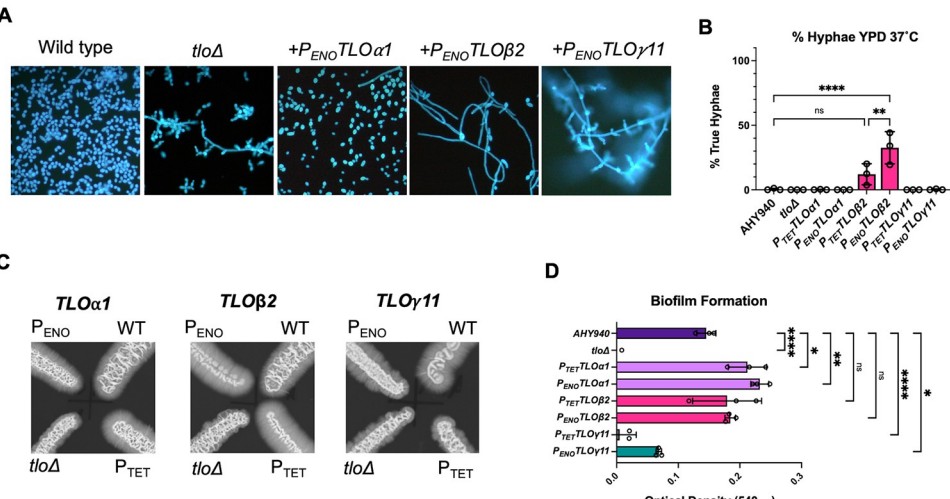

**Fig 6. Complementation of *tloΔ* mutant phenotypes by *TLOα1*, *TLOβ2* or *TLOγ11*.** (A) Morphology of WT, *tloΔ* mutant and *tloΔ* mutant cells complemented with $P_{ENO1}$ expressed *TLOα1*, *TLOβ2* and *TLOγ11* grown in YEPD medium at 37˚C. (B) Percentage true hyphal cells observed in overnight YPD cultures. P values (one-way ANOVA) showing significant difference from WT are indicated as follows: **** = p < 0.0001; ** = p <0.01). (C) Morphology of WT, *tloΔ* mutant and *tloΔ* mutant cells complemented with *TLOα1*, *TLOβ2* and *TLOγ11* grown on Spider medium at 30˚C. (D) Biofilm formation on 96-well tissue culture dishes (Greiner) following incubation in Spider medium at 37˚C. Quantification was carried out with crystal violet as described in the methods. P values (one-way ANOVA with Dunnett's multiple comparisons) showing significant difference from WT are indicated as follows: **** = p < 0.0001; ** = p <0.01, * = p < 0.05).

cells and hyphal cells (Fig 6A) which conferred a distinct wrinkled colony morphology on solid YEPD medium (S12 Fig). Expression of *TLOβ2* from the stronger $P_{ENO1}$ promoter resulted in significantly increased numbers of true hyphal cells in YEPD cultures compared to the $P_{TET}$ regulated gene (Fig 6B). Increased expression of *TLOα1* from the $P_{ENO1}$ promoter did not affect colony or cellular morphology. Integration of *TLOγ11* did not significantly alter the cellular or colony morphology of the *tloΔ* mutant in YEPD medium (Figs 6A and S12). This was confirmed with a second *TLOγ* gene, *TLOγ5*, which was expressed from the $P_{ENO1}$ promoter and also did not alter *tloΔ* mutant morphology on YEPD medium (S13 Fig). On Spider agar, each *TLO* gene-expressing reintegrant conferred a different phenotype: *TLOα1* restored a hyphal fringe similar to WT, *TLOβ2* expression induced a hyphal fringe in excess of the WT phenotype, $P_{TET}$-*TLOγ11* reintegrants phenocopied the *tloΔ* mutant, while $P_{ENO}$-*TLOγ11* resulted in an increased hyphal fringe compared to the mutant phenotype (Fig 6C).

We next examined the capacity of *TLOα1*, *TLOβ2* or *TLOγ11* genes to complement the defect in biofilm formation observed in the *tloΔ* mutant (Fig 6D). The $P_{ENO}$-*TLOγ11* gene could partially restore biofilm formation and the *TLOβ2* constructs restored a phenotype similar to WT, whereas the *TLOα1* constructs ($P_{TET}$ and $P_{ENO1}$ driven) restored biofilm to a significantly greater extent than WT (Fig 6D).

## *TLO* genes differ in their capacity to restore growth rate

In order to investigate if each of the representative *TLO* genes restored WT growth rates, we measured doubling times in media containing glucose and galactose as the sole source of carbon and also the capacity of each gene to restore expression of genes required for metabolism of these sugars.

Complementation of the *tloΔ* mutant with the $P_{TET}$- or $P_{ENO1}$- expressed *TLOα1* genes restored wild-type growth rates in YEP-Glucose (Figs 7A andS14). $P_{TET}$-*TLOα1* yielded

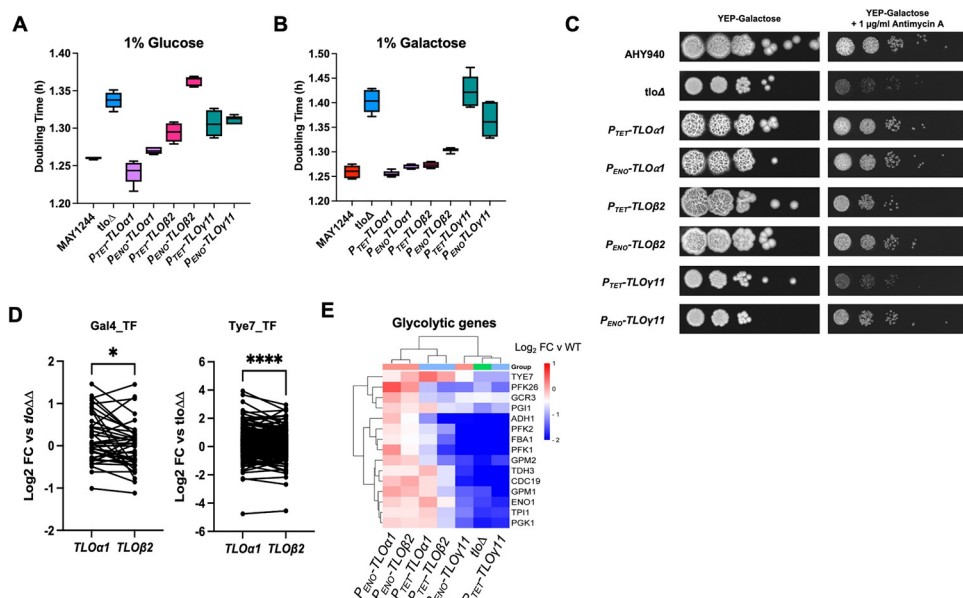

**Fig 7. Growth and gene expression in WT, *tlo*Δ mutant and *tlo*Δ mutant cells complemented with *TLOα1*, *TLOβ2* and *TLOγ11*.** Growth rates in liquid YEP (A) plus 1% glucose (w/v) and (B) 1% galactose (w/v) at 37°C. (C) Growth on solid YEP-Gal and YEP-Gal supplemented with antimycin A. (D) Expression of genes regulated by the transcription factors Gal4 and Tye7 in the P$_{TET}$-*TLOα1*, P$_{TET}$-*TLOβ2* complemented strains relative to the *Δtlo* mutant. Wilcoxon test p values; **** = <0.0001; * = 0.014. (E) Heatmap comparing expression of glycolytic genes in *TLOα1*, *TLOβ2* and *TLOγ11* complemented strains relative to WT.

doubling times faster than WT MAY1244 and at lower glucose concentrations (≤0.5%) this difference was significant (S14 Fig). Complementation of the *tlo*Δ mutant with P$_{TET}$-*TLOβ2* partially complemented the growth defect in YEP-Glucose (Fig 7A). However, complementation of the *tlo*Δ mutant with P$_{ENO}$-*TLOβ2* did not restore WT growth rates, although this may be due to the hyphal nature of growth exhibited by this strain under these culture conditions. Similarly both *TLOα1* and *TLOβ2* could restore growth rates similar to WT in galactose-containing liquid medium (Figs 7B and S15) and on solid medium containing galactose and the respiratory inhibitor antimycin A (Fig 7C). *TLOγ11* could only partially restore WT levels of growth in glucose (Fig 7A), while in galactose-containing medium, P$_{TET}$-*TLOγ11* was unable to restore growth, while P$_{ENO}$-*TLOγ11* restored growth to a limited degree (Fig 7B and 7C). Similarly, P$_{ENO}$-*TLOγ5* could only partially restore growth in glucose or galactose-containing medium (S13 Fig).

Both *TLOα1* and *TLOβ2* significantly affected expression of genes involved in glucose metabolism including genes bound by the transcription factor Tye7p (Fig 4B). Due to the greater capacity of *TLOα1* to restore growth in glucose medium, we compared the expression of these gene sets in *TLOα1* and *TLOβ2* complemented strains. Significantly greater expression of genes bound by Tye7 (p <0.0001) and Gal4 (p = 0.015) was observed in *TLOα1* complemented strains compared to the *TLOβ2* complemented strains (both P$_{TET}$ and P$_{ENO}$ constructs) (Fig 7D). Reintegration of *TLOα1* or *TLOβ2* into the *tlo*Δ mutant resulted in restoration of glycolytic gene expression, although *TLOα1* appeared to act as a stronger activator of gene expression compared to *TLOβ2* and this was most noticeable in comparisons between P$_{TET}$-*TLOα1* and P$_{TET}$- *TLOβ2* (Fig 7E). Increased expression of *TLOβ2* from the *ENO1* promoter resulted in increased levels of glycolytic gene expression which were more equivalent to that observed in Tloα1 reintegrant strains (Fig 7E)

### *TLO* genes show differences in their contribution to stress tolerance

Due to the increased sensitivity of the *tloΔ* mutant to stress conditions (Fig 2), we compared the ability of each of the representative *TLO* genes to complement the stress sensitivity of the *tloΔ* mutant. The reintroduction of each of *TLOα1* or *TLOβ2* into the mutant under either the $P_{TET}$ or $P_{ENO}$ promoter restored tolerance to Congo Red and Calcofluor White (Fig 8A). Under these conditions, *TLOγ11* partly complemented the mutant phenotype (Fig 8A), with high level $P_{ENO}$-driven expression of *TLOγ11* resulting in increased growth in both Congo Red and Calcofluor White relative to the *tloΔ* mutant (Fig 8A). This phenotype was not observed in the second gamma family gene examined, *TLOγ5* (S13 Fig).

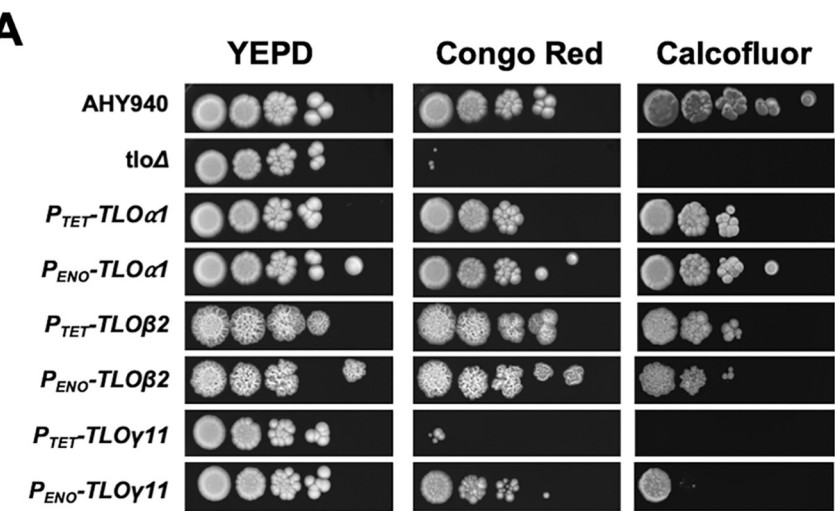

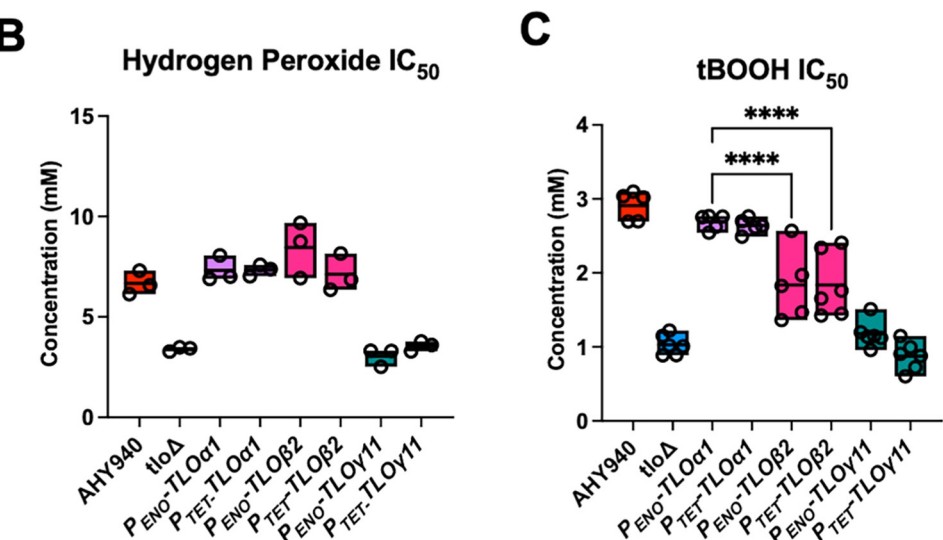

**Fig 8. Susceptibility of *TLO* complemented strains to stress.** (A) Susceptibility of the *tloΔ* mutant and *TLOα1-*, *TLOβ2-* and *TLOγ11*-complemented strains to cell wall stress induced by Congo Red (25 μg/ml) and Calcofluor White (150 μg/ml) incorporated in YEPD agar plates. (B) IC$_{50}$ values of hydrogen peroxide and tBOOH for the *tloΔ* mutant and *TLOα1-*, *TLOβ2-* and *TLOγ11*-complemented strains. P values (one-way ANOVA) showing significant difference between *TLOα1* and *TLOβ2* in (C) are indicated with **** = p < 0.001.

Each of the representative *TLO* genes differed in their capacity to complement the defective expression of oxidative stress response genes, with *TLOα1* reintegrants showing the greatest GSEA enrichment score in this category (XS(2006)_UP; Fig 4B and S10–S13 Tables) and with *TLOγ11* failing to exhibit any significant restoration of expression of oxidative stress response genes. Expression of either *TLOα1* or *TLOβ2* in the *tlo*Δ mutant background restored wild-type tolerance to $H_2O_2$ while expression of *TLOγ11* was unable to restore oxidative stress tolerance (Fig 8B). *TLOα1* and *TLOβ2* were found to differ in their ability to restore resistance to tBOOH. *TLOα1* could restore WT levels of tolerance while *TLOβ2* could only restore a level of tolerance intermediate between the WT and *tlo*Δ mutant, even when expressed from the strong *ENO1* promoter (Fig 8C).

## *TLO* genes show differences in their role in transcriptional responses to serum

Hyphal morphogenesis of the mutant and reintegrant strains in liquid medium was quantified following incubation in YEPD + 10% bovine fetal calf serum (FCS) at 37°C (Fig 9A and 9B). Following transfer to 10% FCS, the *tlo*Δ mutant strain was unable to form true hyphae (Fig 9A and 9C). Under these inducing conditions neither the *TLOγ11* nor *TLOγ5* gene was able to rescue this phenotype (Figs 9C and S13). However, reintroduction of *TLOα1* or *TLOβ2* restored the ability to form true hyphae, with the $P_{ENO}$ expressed genes exhibiting significantly greater numbers of hyphal cells compared to wild-type after 1h (Fig 9A). Strains expressing *TLOβ2* under control of the $P_{ENO}$ promoter exhibited significantly longer germ-tubes after 30 min incubation compared to wild-type and $P_{ENO}$-*TLOα1* (Fig 9B). RNA-seq analysis was used to compare the transcriptomes of the *tlo*Δ mutant and WT MAY1244 following 30 min incubation in YEPD + 10% FCS at 37°C. This identified the induction of a typical hypha-specific transcriptional response in MAY1244 involving 726 transcripts expressed 2-fold or greater (FDR q <0.05; Fig 9D). This included induction of signature hyphal genes including *ALS3*, *ECE1*, *HWP1*, *HYR1* and *UME6* and repression of *HSP30* and *RBR1*. Hierarchical clustering was used to compare the transcriptional responses of MAY1244 and the *tlo*Δ mutant which illustrated a defective response to serum in the *tlo*Δ mutant, with defective induction and repression of the hypha-specific transcriptome (Fig 9D). Complementation of the *tlo*Δ mutant with *TLOγ11* did not restore wild-type hypha-specific gene expression (Fig 9D). However, complementation with *TLOα1* or *TLOβ2* restored a wild type transcriptional response to serum reflecting the production of hyphae (Fig 9D).

## *TLO* genes show differences in their ability to restore virulence in the *Galleria mellonella* infection model

The virulence of the *tlo*Δ mutant and single *TLO* complemented strains was tested using a *G. mellonella* model of systemic disease (Fig 10). These assays found that the *tlo*Δ mutant and *TLOγ11*-expressing strains were significantly less virulent in this model than the WT strain. Reintroduction of *TLOα1* under the $P_{TET}$ promoter, as well as *TLOβ2* under either the $P_{ENO}$ or $P_{TET}$ promoter was able to restore virulence in this model to WT levels, while the reintroduction of *TLOα1* under the $P_{ENO}$ promoter generated a strain which was significantly more virulent than the WT strain in this model (Fig 10).

## Complementation of a *tlo*Δ/*med3*Δ double mutant with *TLOα1*, *TLOβ2* and *TLOγ11*

In order to attempt to investigate if the Tlo proteins require an intact Mediator tail module to function, we created a "double" mutant in which all of the *TLO* genes have been deleted along

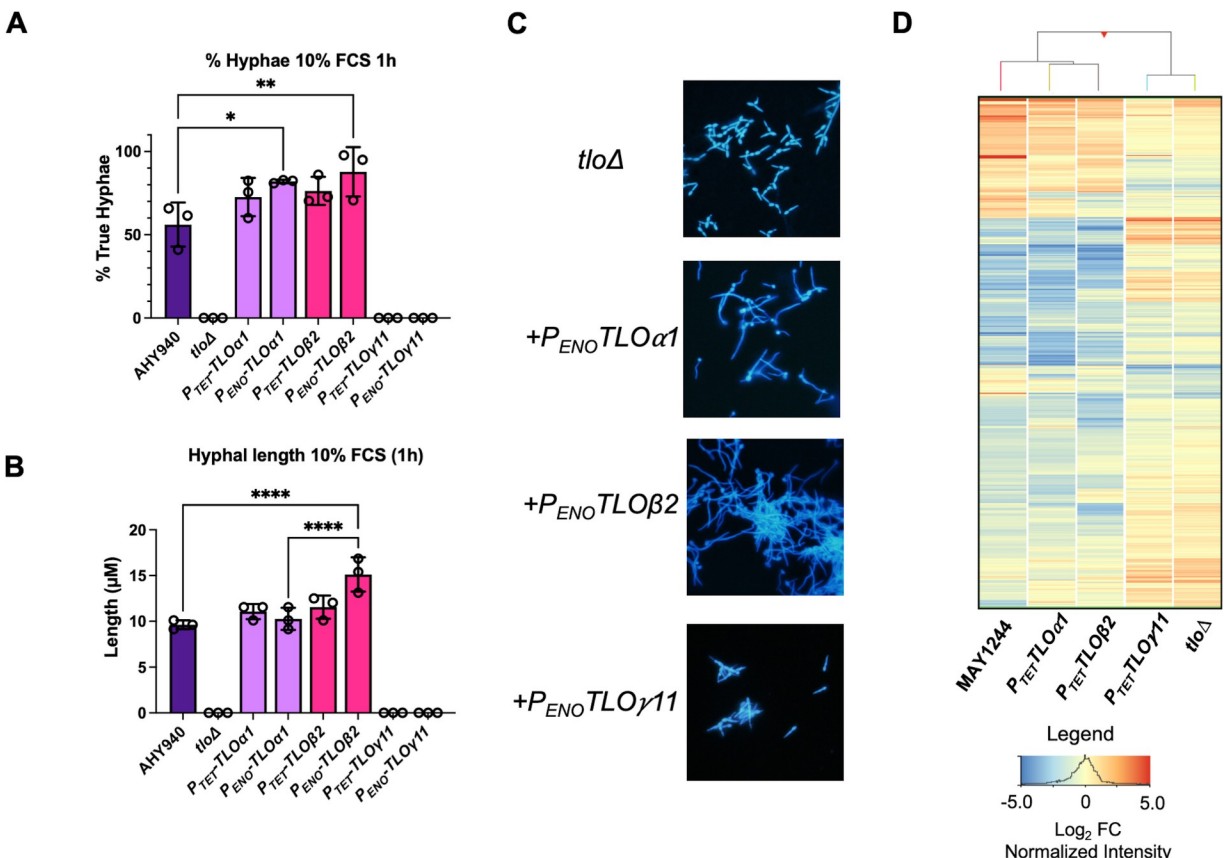

**Fig 9. Analysis of the response of *tloΔ* mutant and *TLOα1*, *TLOβ2* and *TLOγ11* complemented strains to serum.** (A) Percentage true hyphae present in cultures of the indicated strains following incubation in 10% bovine fetal calf serum (FCS) for 1 h at 37°C. (B) Measurements of average hypha length (μM) in 10% FCS following 1 h incubation at 37°C. Significant differences of interest in (A) and (B) are indicated by P values (one-way ANOVA) with **** = p<0.0001; *** = p < 0.001; ** = p < 0.01 and * = p < 0.05. (C) Cellular morphology of the strains in 10% FCS following 2h incubation and calcofluor staining. (D) Heatmap comparing expression levels of genes induced or repressed 2-fold or more in wild-type MAY1244 following 30 minutes incubation in 10% FCS across the *tloΔ* mutant and *TLOα1*, *TLOβ2* and *TLOγ11* complemented strain set. Color indicates Log2 fold-change in expression normalised to the median of all genes.

with the *MED3* gene, which encodes another tail module component which is proposed to anchor Med2/Tlo to the main Mediator complex. This was achieved by deleting the *MED3* gene in the CC16 *tloΔ* mutant using CRISPR-Cas9 mutagenesis. When cell and colony morphology, growth rate and stress resistance were analysed, the *tloΔ*/*med3Δ* double mutant was found to phenocopy the *tloΔ* and med3Δ mutants (S16 Fig). Reintegration of the *TLOα1*, *TLOβ2* and *TLOγ11* genes under the control of the $P_{ENO}$ promoter in the *tloΔ*/*med3Δ* double mutant did not lead to any change in these phenotypes when compared to the *tloΔ* mutant or the parental *tloΔ*/*med3Δ* mutant. This suggests that under these conditions, at least, a functional Mediator tail is required for Tlo proteins to restore wild-type phenotypes.

## Discussion

The extent of the *TLO* gene family expansion in *Candida albicans* is unique compared to other *Candida* species [10], and to other eukaryotes in general. This raises the question: Does this expanded repertoire of Mediator components contribute to the relative success of *C. albicans* as a coloniser and pathogen of humans?

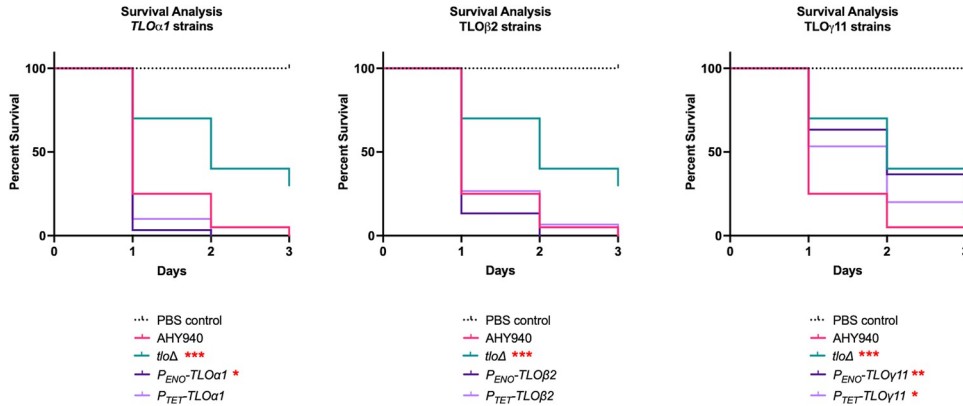

**Fig 10. Survival curves of *G. mellonella* larvae (n = 30) infected with the *tloΔ* mutant and *TLOα1-*, *TLOβ2-* and *TLOγ11-* complemented strains.** Each panel shows the WT (AHY940) and *tloΔ* mutant data sets compared to results obtained with the $P_{TET}$ and $P_{ENO}$ *TLOα1*, *TLOβ2* and *TLOγ11* complemented strains. Results significantly different from AHY940 in a log-rank (Mantel-Cox) test are indicated with red asterisks (* = p <0.05; ** = p 0.01; *** = p 0.0007).

To determine the function of Tlo proteins in *C. albicans*, we applied CRISPR-Cas9 mutagenesis to generate a *tloΔ* mutant, in which all 14 paralogs have been deleted (S1 Fig). This allowed us to investigate our main hypothesis that specific Tlo proteins have diverse functions [28,30,32]. To explore this, we complemented the *tloΔ* mutant with one representative *TLO* gene from each clade (*TLOα1*, *TLOβ2* and *TLOγ11*) and compared the ability of each gene to restore any phenotypic defects associated with the mutant (see Fig 11 for a summary of the ability of each gene to complement the mutant phenotype). Reintroduction of the *TLOα1* or the *TLOβ2* gene into the *tloΔ* mutant background, under either a relatively weak ($P_{TET}$) or a

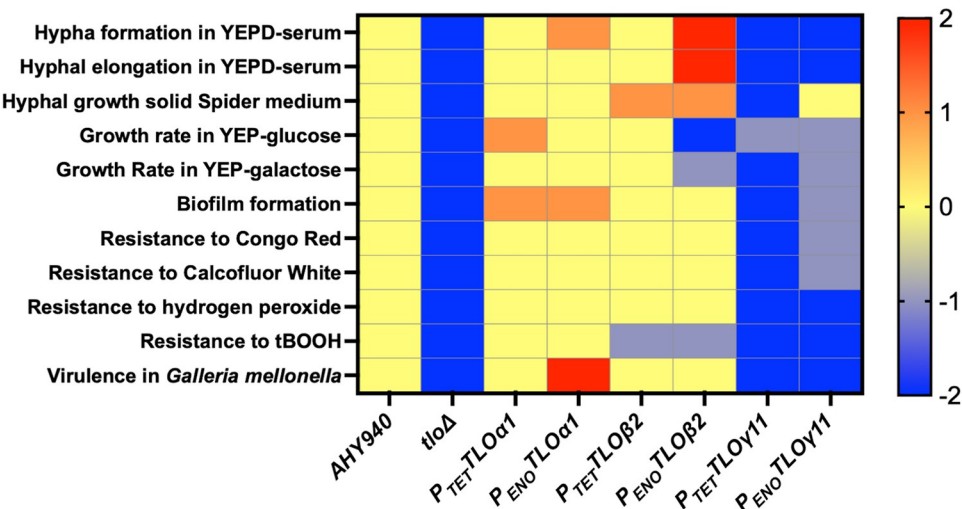

**Fig 11. Summary of *tloΔ* mutant phenotypes complemented by *TLO-α*, *-β* and *-γ* reintegrants.** Wild-type phenotype is indicated in yellow and an arbitrary scale (2 to -2) is used to score mutant phenotypes relative to WT, where the defective *tloΔ* mutant phenotype is -2. Genes that complement an approximate WT phenotype are scored in yellow, partial complementation in grey (-1). Phenotypes in excess of WT and supported by significant p values, where appropriate, are scored +1 or +2, depending on significance. Growth rate assessments are based on data across Figs 7, S14 and S15. The compromised growth of the $P_{ENO}$-*TLOβ2* reintegrant strains in YEP-glucose is likely related to hyphal morphology.

strong ($P_{ENO}$) promoter, restored many of the mutant's phenotypes, such as growth rate, resistance to stress and virulence (Figs 6–11). In contrast, when the same three clade-representative *TLO* genes were expressed from the $P_{ENO}$ promoter in a *tlo*Δ/*med3*Δ double mutant they were unable to complement any of the phenotypes observed, suggesting that the functionality of these Tlo proteins requires them to be anchored to the Mediator complex via Med3 in order to exert their effect, at least under the conditions tested in this study. These data suggest that Mediator-free Tlo has limited functionality or may play roles in phenotypes not examined here. However, in the case of *TLO*β2, some evidence for a Mediator-independent role was identified. Expression of *TLO*β2 from the strong $P_{ENO}$ promoter enhanced filamentous growth and hypha-specific gene expression relative to the $P_{TET}$ expressed gene. Western blot analysis showed that expression of the HA-tagged Tlo proteins from the strong $P_{ENO}$ promoter generated levels of protein ~30-fold higher than HA-tagged Med3, suggesting the presence of a super-stoichiometric, Mediator-free population of Tlo proteins. One interpretation of these data is that the enhanced filamentous growth observed in the $P_{ENO}$-*TLO*β2 reintegrant may be driven by the presence of an excess of Mediator-free Tloβ2 in the cell, similar to the effects described by Liu *et al.* [16] when they generated a Mediator excess pool of Tlo in *C. dubliniensis*. However, this $P_{ENO}$-*TLO*β2-induced phenotype was not observed in the *tlo*Δ/*med3*Δ double mutant, indicating that an intact Mediator tail is still required for this phenotype to manifest. This apparent paradox will require further investigation.

Although Tloβ2 appears to have specific functions involved in regulating hyphal growth and hypha-specific gene expression [33], our data suggest that *TLO*α1 may be a more potent activator of transcription and conferred greater phenotypic changes compared to the *TLO*β2 reintegrant. For example, despite similar protein expression levels, the $P_{TET}$–*TLO*α1 reintegrant consistently demonstrated increased growth rates in YEP-glucose and YEP-galactose compared to $P_{TET}$–*TLO*β2 reintegrants (Fig 7). *TLO*α1-induced enhanced expression of glycolytic genes and higher levels of expression of genes regulated by Tye7 and Gal4. *TLO*α1 also conferred greater resistance to oxidative stress induced by tBOOH and resulted in significantly greater levels of expression of genes required for resistance to oxidative stress under these conditions. However, increasing the expression of *TLO*β2 from the *ENO1* promoter could restore glycolytic gene expression to a similar extent compared to the *TLO*α1-expressing strains, although increased expression of *TLO*β2 did not rescue susceptibility to tBOOH. In terms of virulence, when the three reintegrant strains were tested in the *G. mellonella* infection model, the $P_{ENO}$-*TLO*α1-expressing strain was observed to be the most virulent, exceeding that of the parent WT strain (Fig 10). Transcriptional profiling of *C. albicans* during *G. mellonella* infection previously showed that *TLO*α1 is specifically upregulated in this model, but not during murine infection [34]. Although only one *TLO*α gene was analysed in detail in the current study, the data for *TLO*α1 suggest that this member of the α clade encodes a classical Med2 component with roles in maintaining baseline transcription and enhancing transcriptional responses to stress.

Compared to *TLO*α1 and *TLO*β2, the two *TLO*γ genes analysed in this study (*TLO*γ5 and *TLO*γ11) complemented far fewer of the *tlo*Δ mutant phenotypes did not significantly alter gene expression in the mutant (Fig 11). The two γ-Tlo proteins examined here (Tloγ5 and Tloγ11), were also undetectable by Western blot analysis despite high mRNA abundance. Together, these data suggest that Tloγ proteins (which represent the largest clade) are functionally different compared to Tloα and Tloβ proteins. Gamma-Tlo proteins have previously been detected by Western blotting but at far lower levels than α- and β-Tlo proteins [26]. This may be due to lower translation of the mRNA or protein instability [16]. Although undetectable using Western blotting in our study, we were able to detect Tloγ11 peptides following IP and tandem mass-spectrometry. ChIP-seq analysis also indicated the presence of DNA-bound

Tloγ11 widely across the genome (Fig 5), however the extent of Tloγ11-DNA interactions may be an over-estimate due to the very low levels of Tloγ11 protein present. Although Tloγ11, like Tloα and Tloβ, was found to interact with the RNA Polymerase subunit Rpc10, none of the Tloγ11 replicates demonstrated association with any Mediator tail components. Previously, Zhang *et al.* [13] found direct evidence for Tloα and Tloβ incorporation in Mediator following purification of the complex from WT cells, however no evidence of a Tloγ protein incorporated in Mediator was detected. These combined data suggest that γ-Tlo proteins are either incorporated into Mediator less commonly than members of other Tlo clades, or that they act independently of Mediator, despite the presence of the N-terminal "Med2" domain, which has been suggested to be involved in Mediator incorporation [13,30,35]. The apparent weak impact of *TLOγ11* on the transcriptome (Fig 4) could also be explained by the lack of the extended C-terminal domain found in Tloα and Tloβ proteins which has been shown to possess independent transcriptional activating activity [35].

As the *TLOα* and *TLOβ* genes analysed in this study restored many of the *tloΔ* mutant phenotypes, it is still not clear why *C. albicans* has expanded and maintained a large repertoire of seemingly functionally defective *TLO* gamma paralogs. Maintenance of the *TLOγ* group may be related to functions outside the nucleus (e.g. within mitochondria) as suggested by Anderson *et al.* [26]. However, the association of Tloγ11 with the Rpc10 subunit of RNAP II identified here indicates a potential role in transcriptional regulation in the nucleus. This finding suggests that Tlo proteins with weak transcriptional activating activity, such as Tloγ5 and Tloγ11 could compete with more active Tloα and Tloβ proteins for interactions with RNAP II. This competition could be analogous to "squelching", a process whereby transcription is downregulated by overexpression of a transcription factor which is in competition with other proteins for a limited pool of co-factors [16,25,35]. Alternatively, the low levels of Tloγ protein detected in the current study could suggest that Tloγ protein may not be directly involved in transcriptional regulation and instead it may be the case that the gamma *TLO* genes are maintained to produce regulatory RNAs or to maintain telomere integrity.

## Conclusions and future directions

This work has confirmed the hypothesis that the alpha-, beta- and gamma-Tlo clades represent functionally distinct groups of proteins. Our understanding of how this diverse family interacts and regulates transcription is only just beginning. Emerging technologies such as single-cell and long-read RNA-seq, in combination with mutagenesis, will hopefully allow us to dissect the individual roles of these genes in the life-cycle of *C. albicans*.

## Materials and methods

### Strains and growth conditions

*Candida* strains in this work were routinely grown in Yeast Extract Peptone Dextrose (YEPD) medium (liquid or solid) at 37˚C. *E. coli* strains were grown in Lysogeny Broth (LB) medium at 37˚C. A list of strains used in this work can be found in S1 Table. *C. albicans* AHY940 (*LEU2/leu2Δ*) was used to generate CRISPR-Cas9 mutants using the methods described by Nyguen *et al.* [36]. During these studies, *C. albicans* strain AHY940 was found to be trisomic for chromosome 5. For this reason, we recovered a derivative of AHY940 (MAY1244) which is disomic for chromosome 5 for use as a control in experiments with chromosome 5 disomic mutants of AHY940 (see S1 Text for details). Analysis of the MAY1244 genome by whole-genome sequencing confirmed that apart from chromosome 5 copy number, this derivative was identical to AHY940. S2 Fig shows the relationship between the wild-type and mutant strains used in this study.

## Generation of *C. albicans tloΔ*, *med3Δ* and *tloΔ/med3Δ* mutants using CRISPR-Cas9 mutagenesis

The *Candida albicans* CRISPR Cas 9 system described in Nyguen *et al*. [36] was used to delete *TLO* genes from the AHY940 strain. The conserved Med2 region of the *TLO* genes was selected for the guide RNA (gRNA) sequence to enable deletion of multiple *TLO* homologs in one step (*TLO*-ALL; S2 Table and S1 Fig). The repair template was constructed from two oligonucleotide primers (Deletion_1_Top and Deletion_1_Bottom; S2 Table and S1 Fig) and was designed to have homology to the 5' region encoding the Med2 domain and homology to the 3' end of each *TLO* gene (S2 Fig). The strategy was designed to delete the central portion of each *TLO* open reading frame (ORF). For example, in *TLOα1* (753 bp length) the region from +70 to +636 is deleted (S2 Fig). Transformation into the *C. albicans* strain AHY940 was performed by electroporation [37] and positive transformants selected on YEPD with 200 µg/ml nourseothricin. The cassette was recycled from confirmed mutants using the LEUpOUT system described by Nguyen *et al*. [36]. Mutants were screened by routine PCR using a chromosome arm specific primer and a pan-*TLO* primer (S2 Table and S2 Fig). The successful deletion of each *TLO* was confirmed by generation of a truncated PCR product compared to wild-type reaction products. In the first transformation around 35 colonies were screened before a null mutant was obtained, and in the second approximately 25 colonies were screened, an efficiency rate of 2.85% and 4% respectively. Mean efficiency was 3.3%

The *MED3* gene was also deleted in the WT and *tloΔ* mutant by CRISPR-Cas9 mutagenesis using a primer (MED3X) containing a gRNA homologous to nucleotides +46 to +65 of the *C. albicans MED3* gene (S2 Table). A repair template consisting of two overlapping oligonucleotides primers (MED3_REPF/MED3_REPR) with homology to the 5' and 3' prime ends of the *MED3* ORF were used to introduce a 450 bp deletion in *MED3*. The presence of a homozygous deletion was confirmed by PCR with flanking primers (MED3_UPF/MED3_DNR; S2 Table).

## Whole genome sequencing (WGS) and karyotype analysis

To verify strains were *tlo*-null, genomic DNA (gDNA) was extracted from an overnight culture of MAY1035 (Anderson laboratory stock of AHY940), MAY1140, *tloΔ* CC16, and *tloΔ* CC10 using the MasterPure Yeast DNA Purification Kit (VWR, CAT# 76081–694) with the optional RNase A treatment. gDNA concentrations were quantified using the Qubit dsDNA Broad Range Assay Kit (ThermoFisher). Following quantification, the samples were used to prepare libraries using the NEBNext Ultra II FS DNA Library Preparation Kit (New England BioLabs). Briefly, 500 ng of input gDNA was fragmented to a size of 300–700 bp, followed by dual indexing using Illumina TruSeq adaptors. Fragments were then size selected for final library sizes of 470–800 bp, followed by 3 cycles for PCR enrichment. The library was then sequenced on a MiSeq v3, using 2×300 paired-end reads by the Genomic Services Laboratory (GSL) at Nationwide Children's Hospital (The Ohio State University). Reads were demultiplexed and Illumina adaptors were trimmed by AMSL. Read quality was assessed using FastQC (v0.11.7) [38]. Reads were mapped to *Candida albicans* reference genome Assembly 21 (A21-s02-m09-r10)–obtained March 2, 2021 from the *Candida* Genome Database (CGD) website (http://www.candidagenome.org/download/sequence/C_albicans_SC5314/Assembly21/current/ C_albicans_SC5314_A21_current_chromosomes.fasta.gz)–using Bowtie 2 (v2.2.6–2) [39] with default parameters for paired-end data. Samtools (v0.1.19) [40] was then used to generate.bam files, read sorting, and sample indexing. Read alignment quality was interrogated via visual scanning using IGV (Integrated Genome Viewer, v2.9.2) [41,42]. Heterozygosity and ploidy of strains was then checked by uploading the.fastq files to the Yeast Mapping Analysis Pipeline (Y$_{MAP}$) at the Berman laboratory webhost (http://lovelace.cs.umn.edu/Ymap/) [43], utilizing

the setting for correction of chromosome end bias for generating visualization maps of copy number and loss of heterozygosity.

Deletions were also confirmed via WGS using Oxford Nanopore Technologies MinION sequencing using a 1D ligation sequencing kit (SQK-LSK109) along with a barcoding kit (EXP-NBD103). Albacore (ONT) was used for basecalling, and Porechop [44] was used to demultiplex reads and trim adaptors. The NanoPack tool suite was used to examine read quality [45], specifically NanoQC and NanoPlot, while NanoFilt was used to filter out reads with a q score < 10 or length < 500 bp, as well as to crop 50 bp from the head and tail of each read. To study the genome structure of *tloΔ* mutant we aligned the assemblies against the reference genome of *C. albicans* SC5314 using Mummer v3.23 [46]. Subsequently, Minimap2 v0.2 [47] was used for the alignment of the raw ONT reads against the genome of *C. albicans* SC5314 to study of the read coverage. SAMtools v1.3 (http://samtools.sourceforge.net, [48]) was used for read extraction, sorting and indexing of the ONT reads aligned over the SC5314 assembly. Bedtools v2.26.0 was used for the alignment transformation from bam to fasta format [49]. Finally reads were visualised using SNAPgene Viewer v5.1.4.1 and the break points were determined by BLASTN. Sequence data is available for download from the NCBI sequence read archive, BioProject no. PRJNA962819.

## Pulsed-field gel electrophoresis

Karyotype analysis was performed using pulsed field gel electrophoresis [50]. The karyotype of the strains was analysed by contour-clamped homogeneous electric field (CHEF) electrophoresis. For this, intact yeast chromosomal DNA was prepared as described by [50]. Briefly, cells were grown overnight in YPD and a volume equivalent to an OD600 of 7 was washed in 50 mM EDTA and resuspended in 20 μl of 10 mg/ml Zymolyase 100T (Amsbio) and 300 μl of 1% Low Melt agarose (Biorad) in 100 mM EDTA and allowed to solidify.

Agarose plugs were submerged overnight in a zymolyase reaction buffer (100 mM EDTA, 100 mM Tris, 1% β-mercaptoethanol). Plugs were then washed in 50 mM EDTA and submerged in proteinase K reaction buffer (100 mM EDTA, 0.2% SDS, 1%) for two nights. Finally plugs were washed in 50 mM EDTA and stored at 4°C until used.

Chromosomes were separated on a 1% Megabase agarose gel (Bio-Rad) in 0.5X TBE using a CHEF DRII apparatus with the following conditions: 60-120s switch at 6 V/cm for 12 hours followed by a 120-300s switch at 4.5 V/cm for 26 hours at 14°C. The gel was stained in 0.5x TBE with ethidium bromide (0.5 μg/ml) for 60 minutes and destained in water for 30 minutes. The gel was visualised using a Syngene GBox Chemi XX6 gel imaging system.

## Reintegration of *TLO*s in the *tloΔ* and *tloΔ/med3Δ* mutant background

Representative *TLO* genes (*TLOα1*, *TLOβ2* and *TLOγ11)* were reintroduced into the *tloΔ* and *tloΔ/med3Δ* mutant backgrounds using the doxycycline inducible pNIM1 expression cassette and under the control of the strong *ENO1* promoter. For the latter constructs, a plasmid derivative of pSFS2a was designed for the integration of each *TLO* at the *TLOα34* locus. A 300 bp segment of the *TLOα34* 3' sequence was amplified from the AHY940 genome and ligated via *Sac*II and *Sac*I cleavage sites in the oligonucleotide primers (S2 Table) to a *Sac*I/*Sac*II digested pSFS2a yielding pJess. *TLO* gene constructs were introduced into the *Kpn*I/*Xho*I sites of pJess. Each respective *TLO* gene construct was synthesised by GeneWiz and consisted of the specific *TLO* gene under the *ENO1* promoter, a 3x HA tag at the 3' end of the gene, and the *ENO1* terminator at the end. A 450 bp sequence with homology to the 5' region of the *TLOα34* locus was also positioned upstream of the promoter. These $P_{ENO}$-*TLO-HA* gene constructs were ligated to *Kpn*I/*Xho*I digested pJess which enabled integration of the $P_{ENO}$-*TLO-HA*-

SAT1-flipper fragment into the genome at the *TLOα34* locus (S7 Fig). Plasmids were linearized with *Kpn*I and *Sac*I before transformation into the *C. albicans tlo*Δ background. Similar experiments were also performed to express individual $P_{ENO}$-*TLO* genes in the double *med3/tlo*Δ mutant. Transformants were selected with 200 ug/ml Nourseothricin supplemented on YEPD and confirmed by PCR for the *TLO* construct in the *TLOα34* locus (S7 Fig) and by Western Blot using an anti-HA 12CA5 antibody (Roche Diagnostics). To generate the doxycycline inducible gene in pNIM1 [51], oligonucleotide primers were designed to amplify each tagged *TLO* gene from the respective pSFS2a plasmid, incorporating a *Sal*I restriction site at the 5' end and a *Bgl*II site at the 3' end. The PCR products were digested and ligated to a *Sal*I/*Bgl*II digested pNIM1 cassette (S7 Fig). Cassettes were transformed into the *C. albicans tlo*Δ background and transformants were selected on 200 ug/ml nourseothricin YEPD and confirmed by PCR (S7 Fig) and by Western Blot with the anti-HA 12CA5 antibody (Roche Diagnostics).

C-terminal tagging of Med3 with a 3xHA tag was carried out in the *tlo*Δ reintegrant strain expressing $P_{ENO}$-*TLOα1-HA* and in WT May1244. Tagging was carried out using a DNA fragment amplified from the $P_{ENO}$-*TLOα1-HA-SAT1* cassette with the MED3-HA forward and reverse primer pair (S2 Table). This fragment contained the 3xHA sequence and the SAT1-flipper cassette region flanked by regions with homology to the 3' end of *MED3*.

### *TLO* expression analysis

The expression of the reintegrated *TLO* mRNA was confirmed by qRT-PCR. cDNA was generated from RNA extracted from strains using the RNeasy kit (Qiagen). Specific primers for qRT-PCR can be found in S2 Table. qRT-PCR was carried out in biological triplicate using an Applied Biosystem 7500 Fast Real Time PCR System, with the expression of *ACT1* used as an endogenous control. Differential expression was calculated as described by Schmittgen and Livak [52] and the results graphed in GraphPad Prism (v9). Protein expression was confirmed via Western Blot with anti-HA 12CA5 antibody (Roche Diagnostics). Expression levels were determined by densitometry using GelAnalyzer 19.1 (www.gelanalyzer.com). Background corrected Tlo band intensities were expressed relative to the RNAPII loading control.

### Phenotypic analysis

Growth rate analysis was carried out in YEP-Glucose and in YEP-Galactose at 37˚C. Cultures were standardised to an $OD_{600}$ of 0.1 in 25 ml (inoculated from an overnight culture) and the $OD_{600}$ was measured every 2 hours. Three time-points from the exponential phase of growth in at least three biological replicates were used to calculate the exponential growth rate in GraphPad Prism (v9). Hyphal growth was induced by inoculating cells into YEPD with 10% fetal bovine serum (FBS) to an $OD_{600}$ of approx. 0.2 from cultures grown overnight in YEPD at 30˚C and incubating at 37˚C in either a static or a shaking incubator. Hyphal formation was monitored using a Zeiss AX10 stereomicroscope microscope at 30 min, 1-hour and 3-hour intervals. Spot plate assays, microbroth dilution assays, biofilm formation assays and *G. mellonella* infection model experiments were performed as described in Flanagan *et al.* [27].

### RNA-sequencing and analysis

Overnight cultures were grown from single colonies in 4 ml YEPD at 37˚C with 200 rpm shaking, fresh YEPD was inoculated from the overnight cultures to an $OD_{600}$ = 0.1 and incubated at 37˚C with 200 rpm shaking until an $OD_{600}$ = 0.8 was reached. RNA was then extracted from strains using the RNeasy extraction kit (QIAGEN) per the manufacturer's instructions. mRNA sequencing was performed with strand-specific libraries and sequenced on the Illumina Nova-Seq 6000 Sequencing System using paired end 150 bp reads. Each experiment generated a

minimum > 20 million read pairs per sample with Q30 score ≥ 85%. Raw reads were aligned to the *C. albicans* SC5314 Assembly 21 genome (downloaded from CGD) in the Strand NGS 4.0 software package using the default settings. Read were quantified and normalised in Strand NGS using DeSeq2 [53] and statistical analysis of differential expression was carried out with post-hoc Benjamini-Hochberg testing performed by default (FDR q < 0.05). Further analysis on lists of differentially expressed genes were performed via GO analysis on the *Candida* Genome Database [54–56] and GSEA [57].

Sequence data is available for download from the NCBI sequence read archive, BioProject no. PRJNA962476.

## ChIP-sequencing and analysis

Cells were grown, fixed and spheroplasted as described by Haran *et al.* [28]. DNA digestion was performed using micrococcal nuclease (MNase). Briefly, sphaeroplasts were incubated with 7U MNase at 37°C with 200 rpm shaking for 4 minutes, before quenching digestion with 250 mM EDTA [58]. Cells were kept on ice and centrifugation steps were carried out at 4°C. Gel electrophoresis of an aliquot of each sample allowed for visualization of the digestion products, with optimum digestion displaying strong bands at 150 bp, 300 bp and 450 bp. The immunoprecipitation was carried out as described by Haran *et al.* 2014, using the anti-RNA Polymerase II, CTD, cl8WG16 antibody or the anti-HA 12CA5 antibody (Roche Diagnostics) [28]. DNA purification was performed using a QIAquick PCR Purification Kit (QIAGEN) according to the manufacturer's instructions. Samples were quantified using a Qubit (Thermo Fisher Scientific). Samples were sequenced on the Illumina platform generating 150 bp paired end reads to a depth of 40 to 50 million reads per sample.

Fastq files were run through FastQC software [38] to examine quality. The fastp tool [59] was used for quality filtering and Illumina adaptor trimming. Paired end files were aligned to the reference genome using Bowtie2 [39], followed by Samtools [40] for sorting and indexing. MACS2 [60] was used to call peaks in the IP samples relative to the control input samples (no selection by immunoprecipitation) and statistically significant peaks were selected (FDR q <0.05). Broad peaks were called for the RNAP samples, and narrow peaks for the anti-HA samples. Replicates were analysed for consistency using bedtools intersect, part of the bedtools suite [49]. The MACS2 tools was then rerun to combine the replicate bam files and output a single peak file. Bedtools intersect was used to determine if peaks from the sample files intersected any ORFs/promoters in the reference genome. These lists of genes were then subjected to further analysis, including GO analysis on the *Candida* Genome Database [54–56]. Expression of each gene list was also analysed using corresponding RNA-seq data for each *TLO* gene using GSEA [57]. The deepTools package [61] was used to generate bigwig files for viewing read pileups in Integrated Genome Browser [62]. Sequence data is available for download from the NCBI sequence read archive, BioProject no. PRJNA962549.

## Mass spectrometry

*Candida* strains were grown at 37°C with 200 rpm shaking until an $OD_{600}$ ~0.5 was reached. Cells were washed once following centrifugation at 2,000 x g for 5 min and resuspended in 450 μl of pre-chilled protein extraction buffer (300 mM NaCl, 50 mM Tris-HCl [pH 7.5], 10% Glycerol, 1 mM EDTA, 0.1% NP-40) supplemented with 1 mM PMSF, 5 mM Benzamidine, 1X Protease inhibitor (Roche), 1X Phosphatase inhibitor (1mM NaF, 0.5 mM sodium orthovanadate, 8 mM b-glycerol phosphate). Cells were disrupted by bead-beating with 425–600 μM glass beads (Merck). Tubes were processed in a mini-beadbeater-24 (Biospec Products) set at 3,000 oscillations per s for 30 s, then cooled on ice for 2 mins, this process was repeated a

further 5 times. Lysis was confirmed by staining an aliquot with Trypan blue and examining under a microscope, lysed cells stained blue. After cell lysis, 50 µl 10% Triton-X-100 was added to each sample and placed on a rotating wheel for 10 min at 4˚C. After incubation, tubes were centrifuged at 10,000 rpm x g for 15 min at 4˚C. Supernatant was collected, moved to a new tube and centrifuged again with the same settings. Supernatant was collected, moved to a new tube and used directly for immunoprecipitation (IP).

Proteins were immunoprecipitated using anti-HA antibody bound magnetic beads (Pierce) and subjected to trypsin digestions as described by Frawley and Bayram [63]. All traces of detergent were removed from trypsin digested samples according to the HiPPR detergent removal spin column kit (Thermo Fisher). Upon completion of this protocol, samples were centrifuged at 10,000 x g for 10 minutes and then dried in a SpeedVac for 2 h. Samples were stored at -20˚C until further use. Peptide samples were resuspended in 20 µl resuspension buffer (0.5% TFA) and sonicated for 5 minutes, followed by a brief centrifugation. Digested samples were purified using ZipTip C18 pipette tips (Millipore) prior to mass spectrometric analysis as described by Frawley and Bayram [63]. Immediately prior to loading, peptide samples were resuspended in 15 µl Q-Exactive loading buffer (2% acetonitrile, 0.5% TFA) and 8 µl was added to mass spectrometry vials (VWR). Samples were loaded on a high-resolution quantitative LC-MS Q Exactive mass spectrometer (Thermo Fisher). Samples were loaded by the use of an autosampler onto a $C_{18}$ trap column, which was switched on-line with an analytical BioBasic $C_{18}$ PicoFrit column ($C_{18}$ PepMap; Dionex) (75-µm inside diameter [i.d.] by 500 mm, 2 µm particle size, 100 Å pore size). Full scans in 300 to 1,700 *m/z* were recorded with a resolution of 140.000 (*m/z* 200) with a blocking mass set to 445.12003. LC-MS identifications of peptides were performed using the Proteome Discoverer Daemon 1.4 software (Thermo Fisher) and organism-specific taxon-defined protein databases. To act as controls, anti-HA magnetic beads were added to crude protein extracts from wild type strains. These samples were prepared for mass spectrometry analysis as described for the immunoprecipitated HA fusion proteins. Confirmation of protein interactions and unique peptides were determined by isolating only those that appear in the HA-tagged purifications but do not appear in any of the wild type controls.

## Galleria mellonella *infection model*

Wax moth larvae (*Galleria mellonella)* were purchased from livefoods4u (Northampton, UK), and stored at 15˚C in wood shavings in the dark. Larvae were used within two weeks of arrival. Larvae between 0.2 and 0.3 g were selected for infection. For each different inoculum, 10 larvae were placed in a petri dish with wood shavings and lined with Whatman filter paper. An inoculum of 1 x $10^6$ cells in 20 µl was injected into each larva using a 30G insulin U-100 Micro-Fine syringe (BD). The site of injection was the last left proleg, allowing direct injection into the haemocoel. Infected larvae were then incubated at 30˚C. At 24, 48 and 72 h time points the larvae were counted to determine the number of surviving larvae. Data was analysed using GraphPad Prism and Kaplan-Meier curves generated.

## Supporting information

**S1 Text. Generation of MAY1244 and Analysis of Chromosome structure in the *TLO* mutants.**
(PDF)

**S1 Fig. Diagram of the *TLOα1* gene locus outlining the CRISPR-Cas9 deletion strategy used in the current study.**
(PDF)

**S2 Fig. Lineage of the strains derived in this study (full genotypes in S1 Table).**
(PDF)

**S3 Fig. Karyotypes of *tloΔ* mutants CC10 and CC16.**
(PDF)

**S4 Fig. Visualization of karyotype by contour-clamped homogeneous electric field (CHEF).**
(PDF)

**S5 Fig. Expression of genes bound by RNAP II in AHY940 and the *tloΔ* mutant.**
(PDF)

**S6 Fig. Phenotypes of *med3Δ* and *tloΔ* strains CC16 and CC10 on Spider medium.**
(PDF)

**S7 Fig. Constructs used to complement the *tloΔ* strain CC16.**
(PDF)

**S8 Fig. Analysis of anti-HA immunoprecipitation of HA-tagged Tlo proteins.**
(PDF)

**S9 Fig. Gene expression in $P_{ENO}$-*TLO* complemented strains relative to the WT strain MAY1244.**
(PDF)

**S10 Fig. ChIP analysis of HA tagged Tlo proteins.**
(PDF)

**S11 Fig. Phenotypic complementation in the CC10 *tloΔ* mutant by *TLOα1*.**
(PDF)

**S12 Fig. Morphology of *TLO* complemented strains on YEPD agar.**
(PDF)

**S13 Fig. Comparison of phenotypic complementation by $P_{ENO1}$-*TLOγ5* and $P_{ENO1}$-*TLOγ11*.**
(PDF)

**S14 Fig. Analysis of growth rates in YEP-glucose at 37˚C at 200 rpm.**
(PDF)

**S15 Fig. Analysis of growth rates in YEP-galactose at 37˚C at 200 rpm.**
(PDF)

**S16 Fig. Comparative phenotypic analysis of a *tloΔ*/*med3* double mutant (CC99) complemented with *TLOα1*, *TLOβ2* and *TLOγ11*.**
(PDF)

**S1 Table. Genotypes of strains used in the study.**
(XLSX)

**S2 Table. Oligonucleotides used in the study.**
(XLSX)

**S3 Table. Genes differentially expressed (FDR q <0.05, FC > 2.0) between MAY1244 and *tlo* null mutant (CC16) in YEPD during exponential growth.**
(XLSX)

**S4 Table. Proteins identified in tandem mass spectromerty analysis of Tloα1-HA IP experiments (proteins found in 3 replicate experiments).**
(XLSX)

**S5 Table. Proteins identified in tandem mass spectromerty analysis of Tloβ2-HA IP experiments (proteins found in 3 replicate experiments).**
(XLSX)

**S6 Table. Proteins identified in tandem mass spectromerty analysis of Tloγ11-HA IP experiments (proteins found in 3 replicate experiments).**
(XLSX)

**S7 Table. Genes differentially expressed (FDR q $<$0.05, FC $>$ 2.0) between $P_{ENO}$-TLOα1 complemented mutant and *tlo* null mutant (CC16) in YEPD during exponential growth.**
(XLSX)

**S8 Table. Gene differentially expressed (FDR q $<$0.05, FC $>$ 2.0) between $P_{ENO}$-TLOβ2 complemented mutant and *tlo* null mutant (CC16) in YEPD during exponential growth.**
(XLSX)

**S9 Table. Genes differentially expressed (FDR q $<$0.05, FC $>$ 2.0) between $P_{ENO}$-TLOγ11 complemented mutant and *tlo* null mutant (CC16) in YEPD during exponential growth.**
(XLSX)

**S10 Table. Gene categories positively enriched (FDR q $<$0.1) in GSEA analysis of the $P_{ENO}$-TLOα1 complemented mutant versus the *tlo* null mutant (CC16) in YEPD during exponential growth.**
(XLSX)

**S11 Table. Gene categories positively enriched (FDR q $<$0.1) in GSEA analysis of the $P_{ENO}$-TLOβ2 complemented mutant versus the *tlo* null mutant (CC16) in YEPD during exponential growth.**
(XLSX)

**S12 Table. Gene categories negatively enriched (FDR q $<$0.1) in GSEA analysis of the $P_{ENO}$-TLOα1 complemented mutant versus the *tlo* null mutant (CC16) in YEPD during exponential growth.**
(XLSX)

**S13 Table. Gene categories negatively enriched (FDR q $<$0.1) in GSEA analysis of the $P_{ENO}$-TLOβ2 complemented mutant versus the *tlo* null mutant (CC16) in YEPD during exponential growth.**
(XLSX)

**S14 Table. List of genes with Tloα1 enrichment at ORF or promoter regions (-1 to -500) identifed by MACS2 (FDR q $<$ 0.05).**
(XLSX)

**S15 Table. List of genes with Tloβ2 enrichment at ORF or promoter regions (-1 to -500) identifed by MACS2 (FDR q $<$ 0.05).**
(XLSX)

**S16 Table. List of genes with Tloγ11 enrichment at ORF or promoter regions (-1 to -500) identifed by MACS2 (FDR q < 0.05).**
(XLSX)

## Author Contributions

**Conceptualization:** Derek J. Sullivan, Gary P. Moran.

**Data curation:** Matthew Z. Anderson, Derek J. Sullivan, Gary P. Moran.

**Formal analysis:** Jessica Fletcher, James O'Connor-Moneley, Dean Frawley, Peter R. Flanagan, Leenah Alaalm, Pilar Menendez-Manjon, Samuel Vega Estevez, Shane Hendricks, Andrew L. Woodruff, Alessia Buscaino, Matthew Z. Anderson, Derek J. Sullivan, Gary P. Moran.

**Funding acquisition:** Derek J. Sullivan, Gary P. Moran.

**Investigation:** Jessica Fletcher, James O'Connor-Moneley, Dean Frawley, Peter R. Flanagan, Leenah Alaalm, Pilar Menendez-Manjon, Samuel Vega Estevez, Shane Hendricks, Andrew L. Woodruff.

**Methodology:** Alessia Buscaino.

**Supervision:** Alessia Buscaino, Matthew Z. Anderson, Derek J. Sullivan, Gary P. Moran.

**Visualization:** Jessica Fletcher, James O'Connor-Moneley.

**Writing – original draft:** Jessica Fletcher.

**Writing – review & editing:** James O'Connor-Moneley, Alessia Buscaino, Matthew Z. Anderson, Derek J. Sullivan, Gary P. Moran.

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
