## [Decision Letter · Decision Letter 0]

18 Jul 2023

Dear Dr Moran,

Thank you very much for submitting your Research Article entitled 'Deletion of the Candida albicans TLO gene family using CRISPR-Cas9 mutagenesis allows characterisation of functional differences in α-, β- and γ- TLO gene function.' to PLOS Genetics.

The manuscript was fully evaluated at the editorial level and by independent peer reviewers. The reviewers appreciated the attention to an important problem, but raised some substantial concerns about the current manuscript. Based on the reviews, we will not be able to accept this version of the manuscript, but we would be willing to review a much-revised version. We cannot, of course, promise publication at that time.

If you decide to revise the manuscript for further consideration at PLOS Genetics, please aim to resubmit within the next 60 days, unless it will take extra time to address the concerns of the reviewers, in which case we would appreciate an expected resubmission date by email to plosgenetics@plos.org.

We are sorry that we cannot be more positive about your manuscript at this stage. Please do not hesitate to contact us if you have any concerns or questions.

Yours sincerely,

Anja Forche, PhD

Guest Editor

PLOS Genetics

Eva Stukenbrock

Section Editor

PLOS Genetics

Reviewer's Responses to Questions

**Comments to the Authors:**

Reviewer #1: The manuscript by Fletcher et al. presents the accomplishment of long sought after goal in C. albicans genetics - the complete deletion of all members of the TLO gene family in the fungal pathogen. The largely sub-telomeric location of the TLOs as well as the relatively small variation between TLOs made this a challenging task that had been long pursued, but only recently accomplished by these authors. The careful documentation of the genetics of the parent strains and the different isolates on the tlo-delta strain is an impressive and important for many in this field who will want to use these strains for the variety of important experiments that are now possible. With this new tool the authors explore a vast variety of different visual and genomic phenotypes associated with the deletion of the of the TLOs and the restoration of certain TLO genes from the three previously categorized clades. The vast variety of properties of the TLO genes that are investigated can be viewed as both a strength and a weakness of the manuscript. It is a strength in that each one of the figures can be viewed as a starting point for a new line of investigation that could result in a manuscript of its own. However, it could also be viewed as a weakness, in that none of these experiments completely take full advantage of the opportunity afforded by the tlo-delta strain to reach a more definitive view of what the expansion of TLO family in C. albicans is actually doing. I do not think it is practical to ask the authors to completely reconfigure the paper with new sets of experiments that would give a more complete view of the TLOs since this would delay the reporting of this important resource. However, I do believe that the authors should address some of the points that I raise below in the results and/or discussion sections as limitations of the study. If the authors choose to do an experiment or two in order to address one of these points it would be welcome, but not all together necessary to clarify these points.

Broadly what is missing is any clear definition, or really recognition, of the issue of whether the deletion of the TLOs, or any of the restoration of Tlos, is originating from the reconstitution of a stable functional Tail module of Mediator, or a role of the Tlo beyond being a direct member of the Tail module. Points that the need further elucidation to reach more definitive conclusions are:

1. Figure 1 could benefit from some greater context. It would be helpful to compare the impact of the tlo deletion in C. albicans versus the Med3 mutant that they made. Since med2 and med3 deletions have very similar impacts on gene expression in S. cerevisiae, comparing tlo deletion to med3 would help sort out whether the tlo deletion impact on gene expression/RNAP II enrichment is a largely Mediator driven effect. Also a comparison with the scope of the impact of tlo deletion in C. albicans versus tlo deletion in C. dubliniensis or med2 in S. cerevisiae (published) would be useful in giving an idea as to whether the CaTLOs are functioning mainly as a med2 ortholog or have another function.

2. Also related to Figure 1, I could not find whether there was an analysis of the correlation between regions with lower RNAP II enrichment and the genes whose expression were impacted.

3. While it is appreciated that the authors tried two different promoters to get different expression levels of each of the restored TLO genes (Fig. 3), there are some missing points of analysis. A comparison of the amount of the tagged Tlo protein to the amount of a comparable Mediator subunit such as a tagged Med3 subunit or a tagged Tlo protein in the parent strain would help determine whether the amount of the restored Tlo is sub-stoichiometric, stoichiometric, or super-stoichiometric with regards to the other subunits of Mediator. This would be informative to telling us whether the phenotypes associated with the Tlo restoration are reflective of fully occupying Mediator and/or creating a non-Mediator associated population. This could be assessed even more directly by tagging another module of Mediator and seeing what percentage of the tagged Tlo protein was pulled down with Mediator. This really needs to be done at the protein level since comparing the RT-PCR results to the western (Fig. 3) clearly shows that the RNA levels are a poor predictor of protein levels with the Tlos.

4. It is concerning that in Figure 3, that there are no other Mediator subunits other than the two other tail subunits in the mass spec experiments on the pull downs. Earlier studies have shown that the Tlo proteins are stably incorporated into an intact Mediator complex. I think it is important to show a western blot of the pull down experiment to determine how much of the HA tagged protein is actually being pulled down. It is possible that only a small fraction of unknown functional significance is being pulled down and analyzed in the mass spec experiment. It seems like some follow up experiments are needed to validate the some of the other interactions since some are quite unusual, such as finding a single subunit of RNAP II, but not any of the others. None of the structural studies of Mediator/RNAP II complexes suggest such an interaction with Rpc10 (Rpb12) It is also a bit unusual to find translation initiation/elongation factors with a protein that is largely nuclear in localization.

5. In Figure 4 it is interesting to see the various restoration TLO strains compared to each other and the deletion. If seems that there is a comparison missing though. How do the restorations compare to the parent strain? Is the restoration partial? complete? different? There is a little bit of this analysis in panel 4C with the hypha specific genes.

6. With the all the restoration experiments there is a real missed opportunity to determine whether the restoration phenotype is dependent on the Med2 domain or the activation domain. This could be done by restoring a Tlo with just the Med2 domain.

7. Another missed opportunity in the Tlo ChIP experiment would be to determine whether the Tlo occupancy pattern is just the localization of Mediator or something different. This could be accomplished by tagging one or more other Mediator subunits and assessing their localization. In the absence of this it is hard to interpret the significance of the Tlo localization.

8. In the phenotype analysis in Figure 6 and beyond, do any of the restoration properties of the Tlos depend on an intact tail module? This could be done by making a med3 deletion in the tlo-delta background. This gets to the important questions of how much to the effect of the Tlo is through Mediator.

9. It is unclear whether the difference in the impact of the Tlo-alpha versus the Tlo-beta is an issue of sequence or of expression level. The western blot seems to show that the highest level of expression of beta is lower than the lowest amount of alpha. Hence, it is hard to conclude that the Tlo sequence is the differentiating factor.

Smaller Points the authors may want to address:

1. It has been published that med3 deletion and the med15 deletion do not switch from white to opaque (although over expression of Wor1 can cause them to switch). The inability of the tlo-delta strain to switch is likely a result of a non-functional Mediator Tail.

2. It is not all that remarkable that transcription factors were 'notably absent' from the Tlo pull down mass spec experiments. This seems to be almost universally true with pull downs of co-activators since the abundance of transcription factors is low and their affinity between co-activators and transcription factors also is usually weak.

Reviewer #2: Fletcher et al. present a well-written and thorough manuscript describing their work aimed at addressing the question of why C. albicans has an unusually large TLO gene family when compared to other eukaryotes. Their central premise is that expansion of this gene family has enabled neofunctionalization of the TLO paralogs and provided a mechanism for increased phenotypic plasticity in C. albicans. While the generation of a 14-gene deletion strain and the subsequent karyotyping and genomic analysis is significant and well presented, there are a few concerns that temper enthusiasm for what is otherwise a thorough and well-presented manuscript.

The fist major issue pertains to the significant differences in protein expression between the Tlo complementation strains. Particularly, the Tloγ protein is only detectable via IP-mass spec while the Tloα and Tloβ variants are readily detected by western blotting. Also, there are significant differences in expression even between the Tloα and Tloβ variants, with the PTETTLOα strain expressing similar levels of Tlo protein to the PENOTLOβ counterpart. While the authors do comment on the differences in expression, they largely ignore these differences in their analysis and discussion of the various assays which rely upon these strains. While there are clearly some differences in the phenotypes of the Tloα and Tloβ complemented strains, even when comparing the variants that have similar Tlo protein expression levels, this potential caveat to the presented work is not adequately discussed.

The second major concern pertains to the ChIP-seq assay and is also tied to the differential levels of Tlo protein expression in the complemented strains. The authors conclude that the Tloα, Tloβ and Tloϒ variants have distinct yet partially overlapping binding targets throughout the genome, with Tloϒ showing a unique enrichment within highly repetitive and/or highly expressed loci which are not occupied by the other two Tlo protein families. However, as discussed in more detail below, this observation is quite possibly an artifact of the extremely low protein expression level of Tloϒ, which is likely to lead to false positive peaks of “enrichment” at highly repetitive or highly expressed genomic loci. The authors should provide relevant controls to support their current conclusions or temper their conclusions by including potential alternate hypotheses that account for this distinct possibility.

Additional/detailed comments:

The authors need to comment on the size cutoff that is used to delineate a "large LOH"

It would be helpful to know how many colonies they had to screen to get the full tlo knockout strains (i.e. deletion efficiency).

In the ChIP assay the authors use "Gene" to designate regions including ORFs + 500bps upstream, which they indicate harbor most of the Tlo protein enrichment. However, this designation seems somewhat arbitrary, given that transcriptional start and stop sites have been mapped for C. alb in at least two cell types (wh and op) and some of the genes that are impacted by TLO gene deletion have rather long 5' and 3' UTRs (i.e. WOR1 has a 5' UTR of ~2kb in opaque cells). Furthermore, the authors state that most of the enrichment occurs within ORFs rather than in upstream regions, however it is unclear whether the inclusion of 5' and 3' untranslated regions would impact this analysis. For example, do ORFs with long UTRs also have enrichment within the 5' and/or 3' UTRs, or is enrichment still primarily restricted to the coding region of the gene? Figure 5B presents an interesting case for this point; the 5' UTR of GAL4 is ~400bp long, yet Tloγ11-HA appears to be largely restricted to the GAL4 ORF. In contrast, Tloα1-HA appears to localize exclusively within the GAL4 5^' and 3^' UTR regions. Again, referring to figure 5B, Tloβ2-HA enrichment extends beyond both the 5^' and 3^' ends of TYE7, which has a short 5^' UTR and a longer (~120bp) 3^' UTR. It would be helpful if the authors could present a more thorough analysis of Tlo protein localization, incorporating transcriptional start and stop sites into their analysis. Also, Figure 5B should have some indication of the scale on the X axis.

The observation that the Tlo protein with the weakest expression (Tloγ11-HA) yields the most enriched loci in their ChIP-seq assay, while the most highly expressed Tlo protein (Tloα1) yielded the lowest number of bound genes, is concerning. Typically, one would expect the more highly expressed protein to yield more overall ChIP enrichment and more “clean”, significant peaks of enrichment above background. Furthermore, the authors mention that the Tlo proteins with the highest number of peaks (coincidentally those with the lowest level of protein expression) also showed a greater degree of enrichment at repetitive (i.e. MRS and centromeres) and highly expressed (i.e. tRNA genes) regions of the genome, which are well documented sources of false peaks in ChIP data (see references below). While it is plausible that some of the Tlo proteins do in fact bind to MRS, centromeric, and tRNA coding loci, it is also very likely that the low level of overall enrichment that would be expected from poorly expressed proteins (i.e. Tloγ11-HA) may have led to overamplification of repettitive/higly expressed sequence elements in the genome, while the increased enrichment (at non-repettitive loci) observed with a more abundant protein may have reduced the amount of unintended amplification of these sequences. The addition of an input DNA control and/or mock IP sample would enable the authors to directly assess this question and determine more definitively what loci are truly occupied by which Tlo proteins.

https://www.pnas.org/doi/10.1073/pnas.1316064110

https://www.nature.com/articles/nrg2641

https://www.nature.com/articles/ni.2117

On lines 478-480 the authors state that TLOα1 is a stronger activator of glycolytic gene expression compared to TLOβ2, most notably when comparing the PTET expressed versions. However, it is certainly feasible that this difference in expression is due more to the different expression levels of the two Tlo proteins when expressed from the PTET promoter (see fig 3B) as opposed to a genuine difference in ability to induce glycolytic gene expression. Based on overall expression levels of the Tlo proteins, the more appropriate comparisson would be PTETTLOα1 vs PENOTLOβ2, where the two proteins are expressed at similar levels (see fig 3B). In this case, most of the glycolytic genes in figure 7E are similarly expressed with the exception of GCR3 and PFK26. The authors should comment on the relative expression levels of the Tlo proteins under these different promoters and how that might influence the interpretation of their observations in this section of the manuscript. Also, it is unclear which strains are being compared in Figure 7D; are the TLO genes expressed from the TET or ENO promoters?

Lines 495-496 the authors state that reintroduction of TLOα1 and TLOβ2 into the mutant under either TET or ENO promoters restored wild-type tolerance to Congo Red and Calcofluor White, however this is somewhat misleading and inaccurate. These genes were added back individually, not in tandem, and there were at least two combinations of complemented gene and stressor that did not yield wild-type tolerance. This sentence should be edited accordingly.

The sentence beginning on line 501 refers to GSEA enrichment scores and referes the reader to Figure 4A, but no GSEA enrichment scoring is presented in that figure. The addition of a supplemental figure to directly present the referred to GSEA enrichment scores across the different complemented strains would be helpful.

The sequence of genotypes, along with the orientation of the bar charts, is different between figure 9A and 9B, which makes it confusing to compare these two pannels.

Line 708 refers to the “highly expressed PENO-TLOϒ11 gene”, however the protein expressed from this construct could only be detected by mass spectrometry and did not appear on a western blot that easily detected the alpha and beta variants. As such, this statement could easily be misconstrued by a reader that did not closely evaluate the western blot shown in figure 3B. Later in the discussion the authors clarify that while the mRNA is abundant for TLOϒ11, the level of protein produced is very low in comparisson to the other TLO complementation constructs. While this does address this concern, it still feels like the statement in line 708 as written could easily be misconstrued.

Line 141-142 the authors state that Med3 is required for Tlo incorporation into the Mediator complex, and present data indicating that the strains lacking all TLO genes have reduced virulence in the G. mellonella model, yet do not discuss or contextualize the observation that the med3 deletion strain phenocopies the wild-type strain in this model. This result should be addressed and discussed.

Reviewer #3: Summary

In this manuscript by Fletcher, et al. the authors seek to investigate phenotypic impact of TLO gene function in the human fungal pathogen Candida albicans (Ca). The TLO genes are found at subtelomeric chromosomal regions and have undergone unique expansion (14 family members) in Ca as compared to other Candida species and encode for a subunit of the Mediator complex involved in transcriptional activator binding to RNAII polymerase. The reason for this expansion is not understood, but the authors hypothesized that diversity in TLO function may underlie the adaptability of Ca as a human opportunistic at multiple mucosal niches, as loss of individual TLO genes and heterologous expression have been previously shown to alter basal and pathogenic phenotypic traits. Using a combination of genetic, transcriptomic, and genomic approaches, coupled with routine phenotypic and virulence assays, the authors seek to tackle this outstanding question. The authors cleverly constructed a global TLO knockout (tloΔ) by using CRISPR-Cas9 and gRNA that recognizes a conserved site in all TLO family genes. Despite some confounding effects with this approach (addressed below), they demonstrate that tloΔ exhibited defects in carbohydrate metabolism and increased expression of hypha-associated genes as compared to WT. The tloΔ strain was also hyperfilamentous under yeast-growth conditions and hypersensitive to oxidative and cell wall stress, similar to a med3Δ that is predicted to have similar defects in Mediator tail structure-function; it was also less virulent in a wax worm model.

Much of the manuscript then focuses on restoration of individual representative TLO family members (⍺, β, or ) under varied strength constitutive or inducible promoters to delineate which TLO genes impact phenotypes described above. HA-tagging of Tlo members followed by Chip-Seq was used to identify interacting partners. Despite slight regulatory and phenotypic differences depending on promoter strength and specific Tlo allele introduced, Tlo⍺ and Tloβ appear to control relatively similar sets of genes and predicted pathways and bind to GAL4 and TYE7, explaining why the tloΔ has reduced growth in glucose and galactose. Interestingly, restoration of Tlo did not revert a majority of the phenotypes explored, indicating that this family member is unique.

While there is an abundant amount of supportive data and the experiments appear to be rigorously performed by a team of scientists with significant expertise in Ca genomics, the study is largely a characterization of mutant strains with little detailed mechanistic insight. Moreover, the question as to why TLO genes have undergone such dramatic expansion still remains to be answered, although the authors do hint at some interesting explanatory mechanisms (i.e., “squelching”) that are actively being pursued. Please find more detailed comments below.

Major comments

1. The Discussion is entirely too long and is mostly a repetition of the results with little insight as to their interpretation. It needs to be cut in half and focused on why various family members have altered phenotypes, TLO-related mechanisms that may allow for fine tuning of virulence and commensalism, etc. It will make the work feel more impactful. As of now, it reinforces the feeling that this is a characterization study.

2. This is more of a comment. In constructing tloΔ, the authors realize that such a large number of double strand breaks may lead to unwanted chromosomal aberrations. Indeed, they did observe this in their tloΔ which may complicate data interpretation. However, by using two independent mutants with different abnormalities they have somewhat alleviated this concern. More comments of this in the discussion and alternative approaches may be warranted. That said, I do appreciate the upfront admission by the authors and their careful analysis.

3. Did the authors perform transcriptional profiling on the med3Δ? If so, how does this compare to tloΔ? It could be interesting information to add to the story to delineate Tlo-dependent vs. Med3-dependent responses.

4. Fig. 2: It is somewhat surprising that tloΔ shows increased filamentous growth at 30C in YPD, but reduced filamentous growth on Spider medium. Can the authors provide some commentary on this disparity in the Discussion?

5. The virulence data (e.g., Fig. 2F) is very modest. What happens at longer time points (e.g., d4)? Do the curves converge?

6. Given that TLO genes are at subtelomeric regions, how stable is TLO copy number in a WT strain? How much might this impact strain-to-strain phenotypic variation? While this is not addressable in the current study, it could be included in the Discussion.

7. Regarding the above, what does TLO family member gene expression look like under various conditions? Have the authors mined publicly available existing RNA-Seq datasets to better grasp which family members are independently or coordinately regulated to given stressors?

8. What is the role of the TLO family members? They do not appear to control phenotypes similar to the other members tested. Are all TLO members similarly non-functional?

9. I caution the authors to carefully go over Figure numbers and callouts in the text. There are multiple errors. I’ve tried to identify those that need changed below, but an additional check is advised.

Minor comments

1. Line 194: do the authors means “gene expressed during hyphal growth…”?

2. Line 249: should be Fig. S5B

3. Line 252: should be Fig. 2F

4. Line 461: fix “the_tloΔ”

5. Line 462: incorrect figure cited

6. Figure 7: labels on x-axis do not match in panels A and B.

7. Line 552: should read “hyhphae”.

8. Line 947: change to “spheroplasted”

**Have all data underlying the figures and results presented in the manuscript been provided?**

Reviewer #1: Yes

Reviewer #2: Yes

Reviewer #3: Yes

PLOS authors have the option to publish the peer review history of their article (what does this mean?). If published, this will include your full peer review and any attached files.

Reviewer #1: No

Reviewer #2: No

Reviewer #3: No

---

## [Editor Report · Decision Letter 1]

14 Nov 2023

Dear Dr Moran and Co-authors,

Thank you very much for submitting a very thorough revision of your manuscript entitled 'Deletion of the Candida albicans TLO gene family using CRISPR-Cas9 mutagenesis allows characterisation of functional differences in α-, β- and γ- TLO gene function.' to PLOS Genetics.

We have carefully assessed the reviewers comments and your revised manuscript. Our main concern at this point is the discussion, which needs to be improved/rewritten. A substantial part of it is a repetition of your results with very little discussion. My suggestion would be to significantly reduce result repetition by 1) focusing on one or two main take away points per result section and 2) expand the discussion portion.

1) Provide a detailed description of the changes you have made in the manuscript.

Yours sincerely,

Anja Forche, PhD

Guest Editor

PLOS Genetics

Eva Stukenbrock, PhD

Section Editor

PLOS Genetics

---

## [Editor Report · Decision Letter 2]

22 Nov 2023

Dear Dr Moran and all Co-authors,

We are pleased to inform you that your manuscript entitled "Deletion of the Candida albicans TLO gene family using CRISPR-Cas9 mutagenesis allows characterisation of functional differences in α-, β- and γ- TLO gene function." has been editorially accepted for publication in PLOS Genetics. Congratulations!

Yours sincerely,

Anja Forche, PhD

Guest Editor

PLOS Genetics

Eva Stukenbrock

Section Editor

PLOS Genetics

Comments from the reviewers (if applicable):

**Data Deposition**

http://datadryad.org/submit?journalID=pgenetics&manu=PGENETICS-D-23-00629R2

**Press Queries**

---

## [Editor Report · Acceptance letter]

27 Nov 2023

PGENETICS-D-23-00629R2 

Deletion of the Candida albicans TLO gene family using CRISPR-Cas9 mutagenesis allows characterisation of functional differences in α-, β- and γ- TLO gene function. 

Dear Dr Moran, 

We are pleased to inform you that your manuscript entitled "Deletion of the Candida albicans TLO gene family using CRISPR-Cas9 mutagenesis allows characterisation of functional differences in α-, β- and γ- TLO gene function." has been formally accepted for publication in PLOS Genetics! Your manuscript is now with our production department and you will be notified of the publication date in due course.

With kind regards,

Dorothy Lannert

PLOS Genetics

On behalf of:
